

# An Economical Tunable-Diode Laser Spectrometer for Fast-Response Measurements of Water Vapor in the Atmospheric Boundary Layer

Emily D. Wein[1], Lars E. Kalnajs[2], Darin W. Toohey[1]

[1]Department of Atmospheric and Oceanic Sciences, University of Colorado Boulder, Boulder, Colorado, USA

[2]Laboratory for Atmospheric and Space Physics, University of Colorado Boulder, Boulder, Colorado, USA

*Correspondence to: Emily Wein (Emily.wein@colorado.edu) and Darin Toohey (Toohey@colorado.edu)*

**Abstract.**

The high spatiotemporal variability of water vapor in the atmospheric boundary layer possesses a significant measurement challenge with abundances varying by an order of magnitude over short spatial and temporal scales. Herein, we describe the

design and characterization of an economical and flexible fast-response instrument for measurements of water vapor in the atmospheric boundary layer (ABL). The in-situ method of tunable-diode laser spectroscopy (TDLS) in the short-wave infrared (SWIR) was chosen based on a heritage with previous instruments developed in our laboratory and flown on research aircraft. The instrument is constructed from readily available components and based on low-cost distributed feedback laser diodes (DFB) that enjoy widespread use for high-speed fiber-optic telecommunications. A pair of versatile, high-speed ARM-based

microcontrollers drive the laser and acquire and store data. High precision and reproducibility are obtained by tight temperature regulation of the laser via a miniature commercial proportional integrating (PI) controller. The instrument can be powered by two rechargeable 3.5 V lithium-ion batteries, consumes 2 W of power, weighs under 1 kg, and is comprised of hardware costing less than $3,000. The new TDLS agrees within 2% compared to a laboratory standard and displays a precision of 10 ppm at a sample rate of 10 Hz. The new instrument allows users with little previous experience in instrumentation to acquire

high quality, fast-response observations of water vapor for a variety of applications. These include frequent horizontal and vertical profiling by uncrewed aerial vehicles (UAVs), long-term eddy covariance measurements from fixed and portable flux towers, and routine measurements of humidity from weather stations in remote locations such as the polar ice caps, mountains, and glaciers.

## 1 Introduction

The sources, sinks, and transport of water vapor within the atmospheric boundary layer (ABL) are key components to radiation budgets and meteorology (Trenberth et al., 2005). Water vapor in the ABL displays high spatiotemporal variability due to the complex nature of land-surface interactions that drive sources (Santanello et al., 2018) and clouds and precipitation that drive sinks (Larson et al., 2002). At large scales, boundary layer water vapor mixing ratios vary from 1500 parts per million (ppm)





in the Arctic to 25,000 ppm in the Tropics, and they can range over five orders of magnitude from the surface to the upper

troposphere (Wulfmeyer et al., 2015). On scales of 100 to 1000 m, water vapor can vary by tens of precent because of differences in local land surface, temperature dynamics and wind fields (Fischer et al., 2012; Kiemle et al., 2011; Shivers et al., 2019). Observations of this variability are essential for elucidating the underlying mesoscale meteorological processes and quantifying local-scale (100 m) radiation budgets (Fabry, 2006; Ogunjemiyo et al., 2002). Observations of the ABL and its variability with high spatial and temporal resolution are necessary to resolve outstanding issues related to the prediction of

turbulent and convective processes and their impacts. However, observations have been limited by the relatively high cost of existing instruments and lack of high quality data from more economical ones (Geerts et al., 2018).

Satellite-based remote sensing measurements are too coarse to resolve important variations of water vapor on very small scales (Trent et al., 2018), therefore fast-response in situ and LIDAR-based instruments have become the primary methods for observing water vapor from the surface and mobile platforms for process-oriented studies. The latter (e.g., DIALS and Raman

Lidars), capable of multidimensional measurements with spatiotemporal resolutions of 10 m to 100 m and greater than 1 s (Wulfmeyer et al., 2015), are deployed frequently for profiling the ABL; however, their relatively high cost and operational demands limit their usefulness for more widespread deployment. Alternatively, fast-response in situ instruments have found increasing use in a variety of applications for measurements of small-scale variations in the ABL capturing the smallest and fastest atmospheric variations near the surface where the atmosphere is not well mixed (Geerts et al., 2018). Incorporating high

sampling rates faster than 1 Hz, instruments such as the infrared gas analyzers (IRGAs) that rely on non-dispersive infrared light have come to routinely monitor surface-based fluxes of $H_2O$ and $CO_2$ within ecosystems (Aubinet et al., 2012). To date these instruments, tend to be highly specialized and available from a small number of vendors as research-grade tools for observations from weather stations or flux towers. In addition, they typically cost $20,000 or more and they require frequent maintenance and calibration from the original factory.

At the other end of the cost spectrum are various versions of capacitive humidity sensors that have found frequent use among hobbyists and research scientists for routine measurements from surface weather stations (Muller et al., 2015). These tiny sensors, costing only tens to hundreds of dollars, employ thin-film water-sensitive polymers sandwiched between two electrodes. They have been used in radiosondes for more than 40 years, and they can be accurate to ~0.8 % over a wide range of humidities. Although they are small and relatively inexpensive, they respond slowly to changes in water vapor, and they

exhibit measurement biases that limit their usefulness for high-frequency observations (e.g. Miloshevich et al., 2009, 2004; Segales et al., 2022).

As fast in situ observations of $H_2O$ are essential for numerical weather predictions and for investigations of the evolution of the ABL and its turbulence characteristics (e.g. large eddy simulations) (Petersen, 2016; Helbig et al., 2021), a more economical instrument for fast, high-accuracy measurements of water vapor in the ABL is desirable. Here, we report on the

performance of a new, tunable-diode laser spectrometer (TDLS) capable of fast-response measurements of water vapor in the ABL. Demonstrating high accuracy/precision like research-grade commercial instruments, yet exhibiting low cost and flexibility desired for more routine observations, the instrument bridges the gap between the more expensive, highly accurate





fast-response instruments and the relatively inexpensive, but slower response capacitive instruments. The design is an adaptation of previous TDLS instruments that have a 30-year history of use on research aircraft including the NASA ER-2 (May, 1998; May and Webster, 1993), DC-8 (Hallar et al., 2004; Newell et al., 1996), WB-57F (Davis et al., 2007) and NCAR GV (Dorsi et al., 2014). As in a previous design (Dorsi et al., 2014), it employs a commercial telecommunications fiber-coupled distributed feedback (DFB) laser in a generic package with self-contained thermoelectric coolers (TEC) for precise selection of wavelength and to reduce artifacts due to absorption by water vapor in trapped spaces in complex coupling optics. The instrument is built from commercial, off-the-shelf technology, and it exhibits performance comparable to instruments costing orders of magnitude more. The new design is flexible and simple, allowing for accurate and reliable measurements of water vapor for investigators with little previous experience with research grade instruments, while being easily adaptable to different contexts and other atmospheric species.

Several immediate applications are envisioned for this new instrument. One involves fast-response, open-path observations of water vapor from a small uncrewed aerial vehicle (UAV) such as a hexacopter. While this application has already been explored (e.g. Bärfuss et al., 2023; Pillar-Little et al., 2021; Segales et al., 2020; Varentsov et al., 2023) the available instrumentation have slow response and limited vertical resolution (Segales et al., 2022). The instrument described in this paper would be ideal for obtaining observations over very small scales (e.g., centimeters), including obtaining frequent high-resolution thermodynamic profiles at locations where observational gaps limit numerical weather prediction and climate modelling (Kämpfer, 2013). Another application is tracking water-resource loss from reservoirs with ground-based flux measurements. There is a need to increase the density of measurements on specific reservoirs to map out the effect of terrain and variable field inhomogeneity (Friedrich et al., 2018). Expanding sensor networks with an economical instrument that maintains high accuracy and precision to monitor evaporation in regions of complex terrain where there is a need for simultaneous observations can open up new areas of study and fill gaps where there is limited knowledge of the importance of evaporation to water availability, especially in arid regions (Roth and Blanken, 2023). Such a capability will also enable new studies of ecosystem exchange in geographic regions that have been historically underserved, for example in developing countries (Markwitz and Siebicke, 2019; Kim et al., 2022).

## 2 Instrument Design

### 2.1 Hardware Description

The TDLS instrument described here is based off a design reported previously for measurements of condensed water contents from research aircraft (Dorsi et al., 2014); a schematic of which is shown in Fig. 1. A DFB laser diode emitting radiation with a wavelength centered at 1368.6 nm at room temperature (NLK1E56AA, NTT Innovative Devices, Yokohama, Japan) rapidly scans over a strong water vapor absorption line. To avoid damping of high-frequency variations a short (~20 cm), open-path, single-pass optical cell was constructed of low-cost commercial components. Water vapor mixing ratios in the range 2000-20,000 ppm are readily retrieved with high precision (+/- 10 ppm). The primary novelty of the new TDLS is a low-



power, low-cost electronics package that simultaneously drives the laser with rapid linear current ramps over a highly stable wavelength range while acquiring data for subsequent processing of the scans into accurate mixing ratios based on laboratory calibrations.

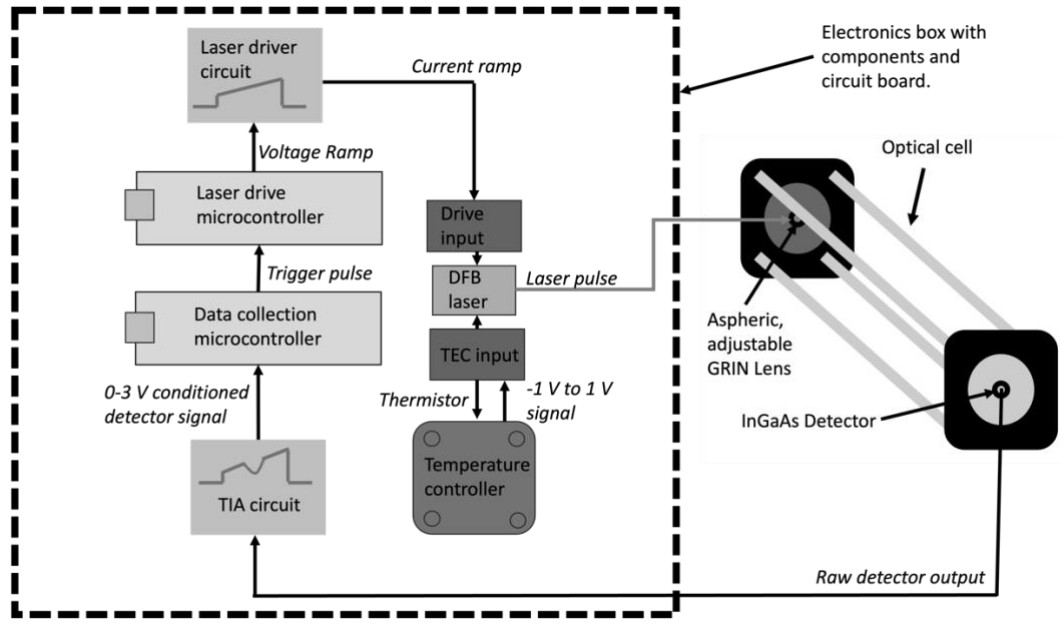

**Figure 1.** Schematic diagram of the new TDLS. Arrows represent direction of information flow between components.


The laser temperature is tuned to the wavelength of a strong water absorption feature centered at 1368.59 nm with a commercial proportional-integral (PI) controller (WTC 3243, Wavelength Electronics, Bozeman, MT). (Gordon et al., 2022).

A highly stable temperature of 0.002 K, consistent with the manufacturer's specification, is maintained with a fixed set point from a voltage divider sourced with a high-precision reference (e.g., LDLN025M25R, STMicroelectronics, Geneva, Switzerland) and a variable resistor. This stability is important for maintaining a reproducible output wavelength of the DFB. If desired, a voltage from the digital-to-analog (DAC) output from one of the microcontrollers can also be used for dynamic temperature control.

Two Teensy Arduino compatible microcontrollers are chosen for the laser-driving ("driver") and data-acquisition ("receiver) functions. These microcontrollers are based off low-cost ARM RISC Cortex-M processors, exhibiting a balance of speed and configurability. Previous instruments employed single or multi-core general purpose processors running full operating systems such as Linux on a PC-104 form factor single board computer (e.g Hallar et al., 2004; Dorsi et al., 2014). Unpublished work in our lab (Rainwater, 2022) showed that imprecise timing of the output ramp for the laser caused by

software interrupts produced unstable PI temperature of the DFB TEC that result in wavelength "jitter" (movement of the



position of the line center in the laser scan). Separating the input and output functions allows for precise control of the laser and highly reproducible scans to ~10 kHz and faster, resulting in high precision of the measurements. The microcontrollers simplify the electronics while also allowing for uninterrupted laser scanning while the detector signal is acquired, processed, and stored.

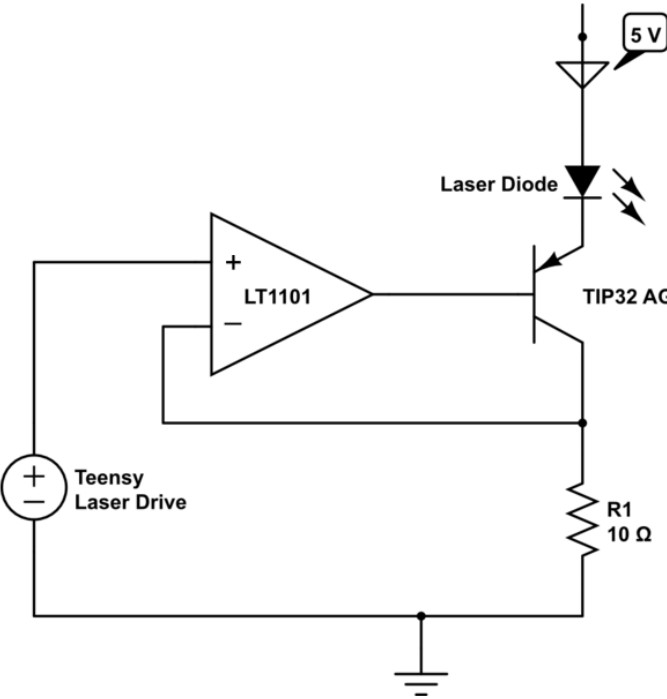


**Figure 2.** Schematic figure for the laser drive circuit described below. The individual components used in this instrument are labeled. R1 requires a resister with a high current tolerance with the current pulled through the transistor contingent on the value of R1 and max voltage output of the Teensy drive.

A Teensy 3.6 with integrated 12-bit, $10^6$ samples per second DAC provides the drive voltage for scanning the DFB current. Prior to the start of each scan, the laser driver produces a pulse ("trigger"), shown on the bottom panel of Fig. 3, on a separate digital line that initiates the data acquisition and storage process on the receiver. The current required to scan the laser is produced by passing the DAC output from the Teensy 3.6 through a custom-built voltage-to-current operational amplifier with the laser diode biased by 5 V. The scan rate, range, and repetition rate period are configured in software. Teensy

model 4.1 was used for data acquisition and storage with a built in Micro-SD card feature. Upon receiving the trigger pulse, the internal clock is recorded into a buffer and the ADC is started. The top panel in Fig. 3 shows the acquired detector output consisting of 445 discreet points (7.2 kHz raw ADC rate) sampled at 12-bit resolution, oversampled 32 times using a built-in averaging function to reduce noise inherent in the Teensy. This results in a minimum resolvable signal of ~0.2 mV. The middle





panel in Fig. 3 shows an example of a series of linear ramps, each consisting of 1366 discreet one-bit steps from 0.80 V to 1.9

V.

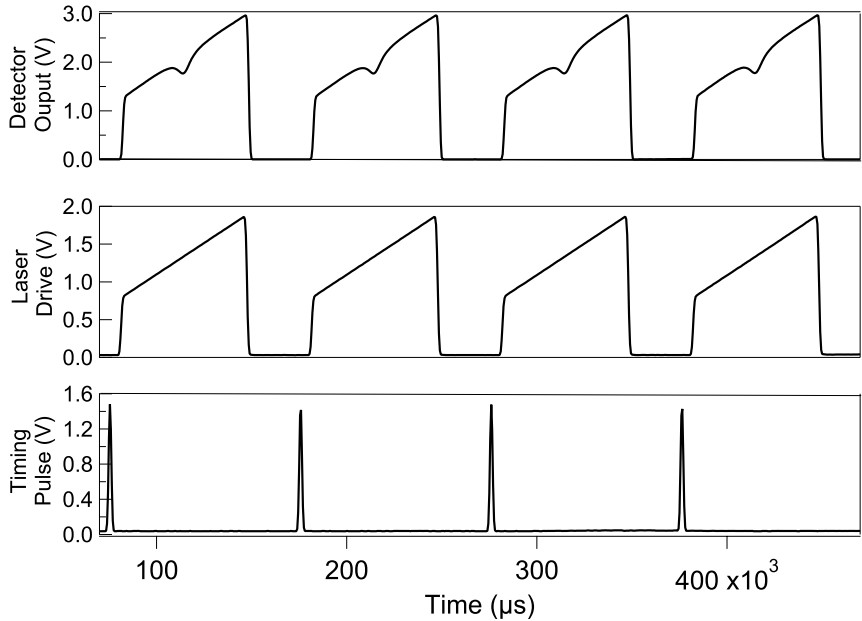

**Figure 3.** Important components of the TDLS laser scans as a function of time. Detector signal (top panel), laser drive voltage (middle panel), and trigger pulse signal (bottom panel). See text for explanation.

For this work, a single pass, open-path, 21.5 cm optical cell was constructed with a fixed 30-mm cage plate assembly (Thorlabs, Newton, NJ). One end housed an adjustable aspheric collimating lens (CFC11A-C Adjustable Fiber Collimator, FC/APC, f = 11.0 mm, 1050 - 1620 nm AR, Thorlabs, Newton, NJ) that was attached to the FC/APC output of the DFB laser, the divergence of which was opened to fully illuminate the active area of a low-noise broadband indium gallium arsenide (InGaAs) semiconductor photodiode (either Thorlabs FDGA05 or Fermionics FD1500) on the opposite side of optical path

both operated in photovoltaic mode. Intensity variations due to vibration and turbulent fluctuations of air density in the optical path are minimized as the beam width is larger than the active area of the photodiode. The photodiode is operated in photovoltaic mode and the photocurrent is converted to a voltage up to a maximum of 3.3 V with a custom-built low-noise transimpedance amplifier circuit using a single-supply operational amplifier amp (AD1101, Analog Devices, Wilmington, MA). The gain could be tuned using a variable resistor, was adjustable with a 1-10 kΩ variable resistor.

The two microcontrollers, laser temperature controller, detector amplifier, batteries, and power conditioning were placed on a custom-built circuit board (OSHPark, Portland, OR). The instrument is powered on or off with a single-pole-single-throw toggle switch, with a small light-emitting diode (LED) that indicates when the instrument is running. An LED on the receiver Teensy indicates when data are being written to the MicroSD card. The entire system consumes 2.5 W, and it can operate for 2 h when powered by two 3.6 V rechargeable lithium-ion batteries (e.g., ARB-L16-700UP, Fenix Lighting, Littleton, CO).





Alternatively, the instrument can be run indefinitely via 5 V passed through the Teensy microUSB input. All components, except the optical cell and coupling laser fiber-optic cable and twisted-pair electrical wires to the detector, can fit in a box with dimensions of 16.18 x 11.18 x 4.90 cm (PN-1324-C, Solutions Direct, Riverside, CA), with the laser output fiber and twisted pair of wires from the detector passing through a hermetic seal.

**2.2 Spectral Processing**

Water vapor concentrations are derived using the approach described previously (Dorsi et al., 2014). Figure 4a shows a single scan over the absorption line consisting of 445 individual measurements of the amplified detector signal. Briefly, a small detector/amplifier offset is determined from 30 points at the start and 20 from the end of each scan while the laser is powered off. Then, linear segments near the beginning and end of the linear current ramp outside of the water vapor absorption

feature are identified for calculating background (i.e., $I_0(t)$) based on a 1st-order polynomial fit (dashed line in Fig. 4a). Scan step number (or elapsed time) is converted to wavelength based on an empirical function derived by using the spacing of line centers of a pair of water absorption lines as a ruler when mapping the laser tune range as a function of laser TEC temperature. This is convenient as it accounts for the possible drift of the tune temperature by removing the nonlinear output laser wavelength in response to a linear current ramp and allowed us to determine the width of the scan to be 0.279 nm. The observed

signal (i.e., $I_{obs}(t)$) and calculated background $I_0(t)$ are then placed in an array *[$\lambda_i$, $I_o(\lambda_i)$, $I(\lambda_i)$]*. Based on the Beer-Lambert Law, water concentration is proportional to the integral of absorbance $A = ln(I_o/I)$ over the full width of the absorption line. This integral is estimated as the sum of discreet points as in Eq. (1).

$$\int A(\lambda) \, d\lambda \; = \; \sum_{k=1}^{385} A(\lambda)_k * \Delta\lambda_k \tag{1}$$


An example of a single laser scan converted to absorbance is shown in Fig. 4b. The resulting integral is related to concentration of water vapor by a response factor determined by laboratory calibration using a high-accuracy cavity ringdown spectrometer, CRDS (L-2120i, Picarro, Santa Clara, CA), referenced to a dew-point generator (LI-610, LiCor, Lincoln, NE) (Noone et al., 2011; Henze et al., 2023). Ambient water concentrations and mixing ratios are interchangeable through the Ideal

Gas Law using concurrent measurements of temperature and pressure, which, for this work, were measured with a small sensor (BMP280, Bosch Sensortec, Reutlingen, Germany) placed midway between the output lens of the laser and the detector just outside the laser beam. The precision of this sensor was measured to be +/-1 Pa and +/-0.01° C.

For this work, we store the raw scan data with T, P and a timestamp and perform data analysis in post processing using code written in Python. This maximizes precision and flexibility while allowing us to evaluate performance with various

diagnostic variables (e.g. those investigating stability or interference) that are only derivable from raw scans. Future iterations of this design will be simplified to include real time processing of the spectra on the Teensy 4.1 before data are written on the microSD card. These calculations take a fraction of the clock cycles needed for writing an entire raw scan so don't affect





instrument time response. In the meantime, we have uploaded our Arduino sketches and processing codes to GitHub opensource.

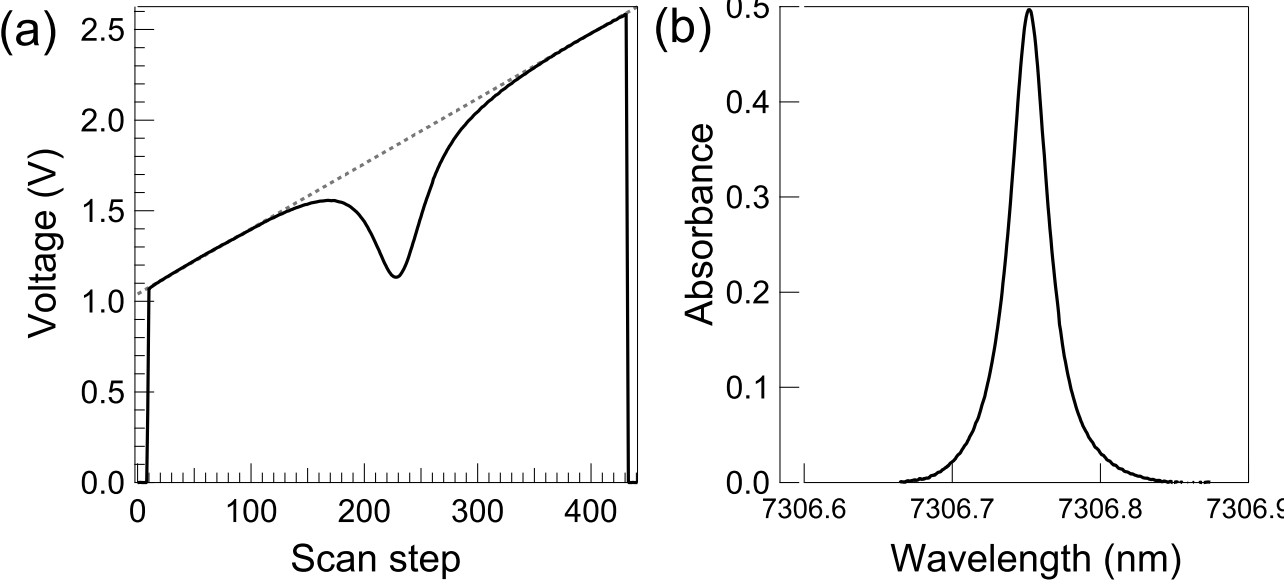


**Figure 4.** (a) Example of the output of the transimpedance amplifier for a single scan of the DFB laser consisting of 445 discreet points. The initial ~10 points and final ~10 points represent the signal with the laser powered off. The dashed line is a linear fit in a region where absorbance by $H_2O$ is negligible (defined as $I_o$). (b) Absorbance defined as $\ln(I_o/I)$ for a single scan of the DFB laser shown in Fig. 3. Wavelength is determined as described in the text.


### 3 Results

The TDLS integrals were calibrated by sampling a series of mixing ratios spanning the range 5,000 ppm to 27,000 ppm in a 250-L Polycarbonate chamber alongside the CRDS. The TDLS optical cell was placed in the center of the chamber, and a fan was used to assure the chamber was well mixed. The sampling line of the CRDS was aligned with the mid-point of the

TDLS open-path cell and positioned just outside the path of the laser beam. The chamber was first saturated to a mixing ratio of ~27,000 ppm with the dew point generator, after which lab air with ~13,000 ppm of $H_2O$ was admitted to the chamber stepwise approximately every five minutes over the course of several hours. Thus, a series of values spanning the range 13,000 ppm to 27,000 ppm was obtained. Values lower than 13,000 ppm were produced by further dilutions using a flow of dry air from a cylinder of Ultra Zero Air (H2O < 2 ppm, total hydrocarbons < 0.1 ppm, Airgas, Dacono, CO). TDLS concentrations

were converted to mixing ratios using pressure and temperature as measured from the BMP280 sensor, and the results are shown in Fig. 5. The deviation between the two data sets is less than 2 % over the range 5000 ppm to 25,000 ppm.

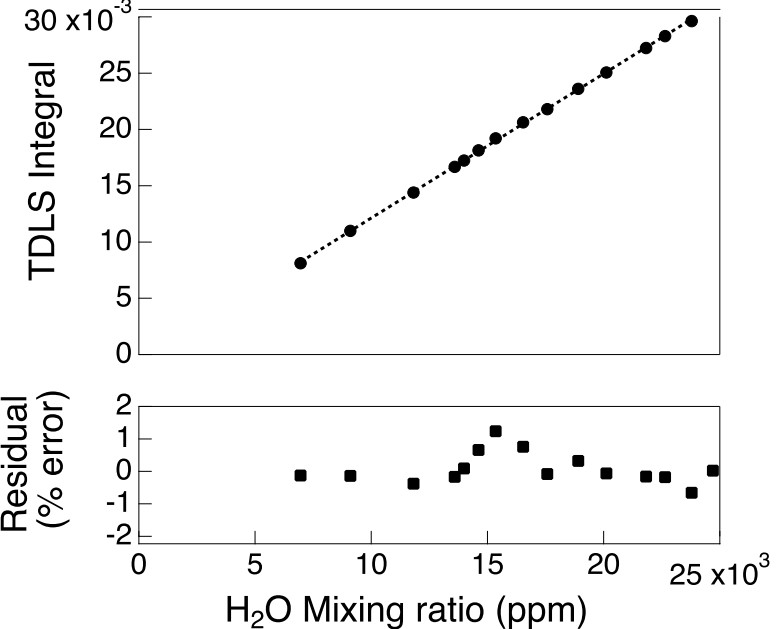

**Figure 5.** Top: Integral signal of the TDLS calculated as described in the text as a function of water vapor mixing ratio (black points). determined by simultaneous measurements with a Picarro L-2120i cavity ringdown spectrometer (e.g., Noone, et. al, 2008). The dashed line represents a linear fit to the results over the range 7,000 – 29,000 ppm. Bottom: Residual error, as percent of measurement, plotted for each of the points in the top panel.

The precision and stability of the TDLS were assessed using a standard Allan-Variance analysis (Werle et al., 1993). Precision is taken to be the square root of the Allan-Variance at the highest sample rate. To reduce variations in ambient water vapor, the output fiber of the laser was attached to one end of the 53.3-cm long sample cell of the CU second-generation closed-path laser hygrometer (CLH-2) that was held at fixed pressure and temperature. The signal was detected with a InGaAs FC/APC-coupled detector (ThorLabs FGA04) as described elsewhere (Dorsi, et al., 2014). This detector is distinct from the ones previously described in this paper and it was used only for this experiment to couple the instrument to the sample cell. In this manner, electronic noise and drift could be assessed independent of variations in pressure, temperature, and water concentration. An Allan-variance analysis of results, shown in Fig. 6, demonstrates a precision of 10 ppm at 0.1-s response time for a water abundance of 11,800 ppm. This represents a fractional absorbance of $10^{-3}$ for the conditions of the test. Averaging (increased integration time) allows the sensitivity to be improved by an order of magnitude down to 0.9 ppm at 34 s corresponding to a sensitivity of 1 in $10^{-4}$.





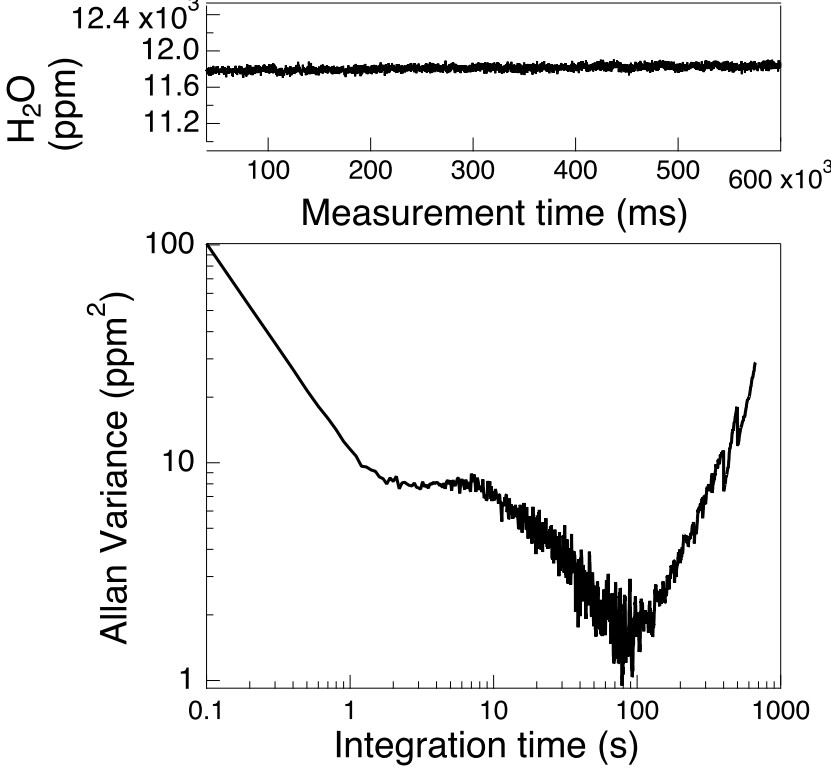

**Figure 6.** Top: Time series of water vapor mixing ratio for a 10-min segment from a laboratory measurement in a sealed absorption cell held at constant temperature and pressure. Bottom: Allan variance calculated from segment of data displayed in the top panel. The instrument demonstrates a precision of 10 ppm at 10 Hz (the intercept in the bottom panel).

The performance of the TDLS was assessed in several "real world" demonstrations, the goals of which were to demonstrate stability for long-term observations and accurate quantification of fast variations of water vapor. The first demonstration was an intercomparison with a commercial analyzer with a long history of eddy covariance measurements of $CO_2$ and $H_2O$ in a variety of environments (e.g. Burns et al., 2009; Ocheltree & Loescher, 2007; Pokorný et al., 2012; Zhao & Tans, 2006). The LI-7000 (LiCor, Lincoln, NE) is a high-performance, dual-cell nondispersive infrared (NDIR) instrument with an accuracy for $H_2O$ of +/- 1% and a precision (RMS noise) of 2 ppm of at 5 Hz (LI-7000 $CO_2$ /$H_2O$ instruction manual; Publication 984-07364, 2007). The site chosen for this test was the exterior of our laboratory where large variations in $H_2O$ would be expected from local sources such as vegetation and passing pedestrians. Figure 7 shows the power series densities (PSD) for both instruments for a 1000-s segment of data.

At frequencies up to ~2 Hz the two instruments exhibit similar behavior, with power dropping with increasing frequency following a −2/3 power law typical for long lived atmospheric variations (Wu et al., 2015). Above 2 Hz, the Li-7000 power spectrum deviates below this power law due to damping of higher frequencies characteristic of closed-path





measurements (e.g., see Aslan et al., 2021). Conversely, the power spectrum of the TDLS trends above the power law at > 3 Hz, exhibiting a measurement precision of ~$10^{-3}$ absorbance, consistent with that determined from the Allan-variance analysis in the static cell, shown in Fig. 6.

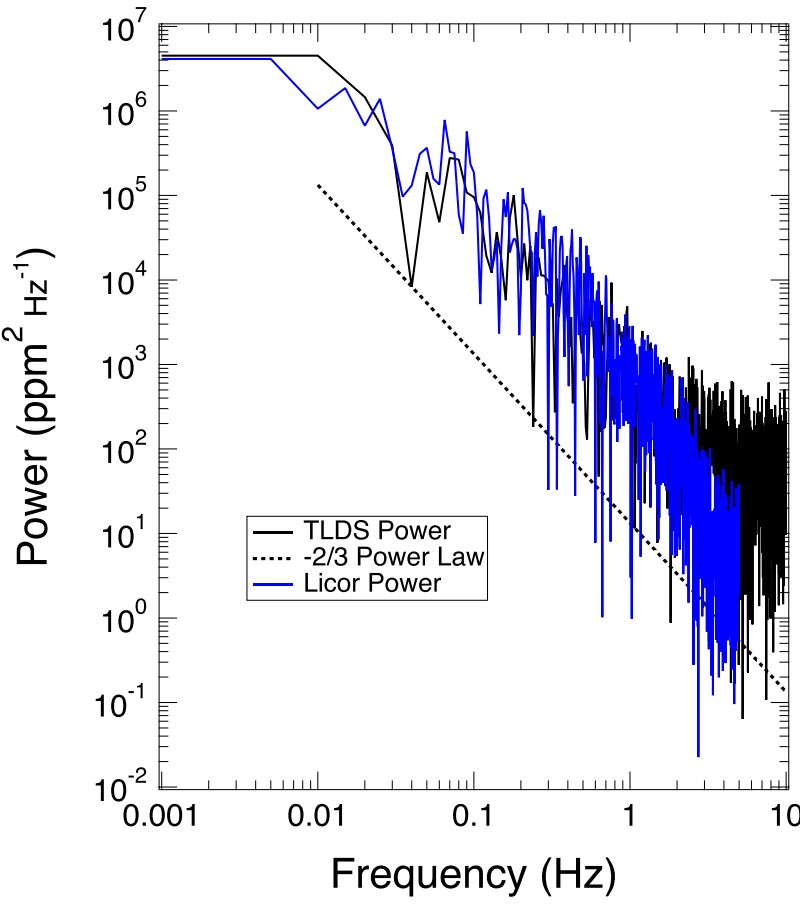

**Figure 7:** Power spectral density (PSD) of the Li-7000 and new TDLS as a function of measurement frequency. The dotted line is the -2/3 power law that is expected for variability of ambient $H_2O$. The Li-7000 PSD does not extend beyond 5 Hz, the maximum sample rate of the instrument.

To test the long-term stability of the TLDS (e.g., days), we performed a three-day intercomparison with the same L2120-i CRDS used for calibrating the TDLS as described above. The TDLS and CRDS sampled from the top of a shipping container used for housing electronics in the Department of Atmospheric and Oceanic Sciences (ATOC) Skywatch Observatory located on East Campus (Lat 40.01° N 105.24° W, Elev: 1600 m) on the University of Colorado at Boulder. The CRDS sampled from a 3-m long, ¼-in O.D. copper line running vertically up the side of the shipping container and terminating with a 3.8-cm radius,





180° bend to avoid ingesting precipitation. The optical cell for the TLDS was installed at the same elevation approximately ~1.5 m from this inlet. A long electrical line and A 25-m long fiber optic patch cable connected the output of the TDLS laser to the collimating lens on the input of the optical cell, and a 10-m long twisted pair brought the detector signal back to the TDLS electronics box which was housed in the shipping container. It is important to note that a better design would have placed the detector amplifier close to the detector to reduce noise pick-up; therefore, this set-up likely represents "worst case"

noise of the TDLS for such a remote installation.

Observations from the TDLS and the CRDS instruments at their native resolutions of 10 Hz and 0.55 Hz, respectively, are shown for three continuous days in Fig 8a. Over this period, $H_2O$ mixing ratios varied from 5,000 ppm to 12,000 ppm, and ambient temperature varied from 10.5 °C to 33.5 °C. There were multiple occurrences of precipitation and virga and periods of variable cloud cover and direct sun. There were several important outcomes from this test. First, the detector/amplifier zero

signal from the TDLS (not shown here) varied from 0.006 V to 0.26 V (i.e., <10 % of average laser signal), from direct sunlight or reflections, thus providing a good test of the validity of the method described above for extracting water vapor mixing ratios from individual spectra. The background was successfully subtracted out before calculation, but this issue could be readily addressed in a proper field experiment by suitable baffling of the optics to block the incoming solar radiation. Second, the robustness and reliability of the spectroscopic foundation of the measurement was demonstrated by successful acquisition of

$4.17 \times 10^6$ unique and independent spectra over this period, with rejection of fewer than 0.05 % due to detector signal that was clipped or filtered when the scan background used to calculate $I_o$ varied by more than 2 %. These losses of signal, which typically lasted only a few seconds and self-corrected, occurred during precipitation on May 4th. They were likely due to condensed water blocking the light path.

A scatterplot of the 3 days of continuous measurements from the TDLS and CRDS is shown in Fig. 8b. Over 5000

observations of 30-second averages are represented in this plot. The TDLS measurements were first averaged in bins of 20 measurements (e.g., to a 2-s time base) and the results then merged to the matching times recorded by the CDRS. Both observations were then bin-averaged down to ~30 s to correspond with the digital smoothing inherent in the Picarro L-2120i instruments. Despite being separated horizontally by ~1.5 m, the two instruments show remarkable agreement over the entire sampling period, with < 4% deviation from a 1:1 correspondence and a 0.993 coefficient of determination ($R^2$). It is noteworthy

that this averaging has removed 80 % of the variability of ambient $H_2O$ largely due to what is occurring on the fastest timescales.



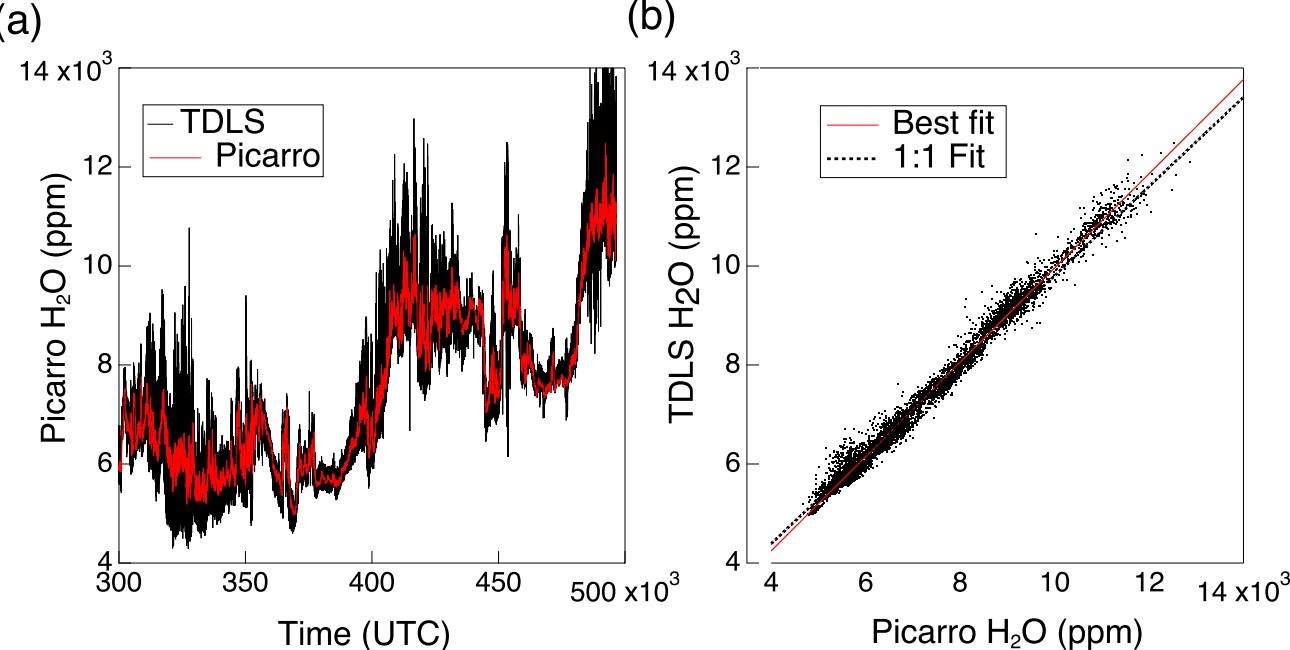

**Figure 8:** (a) Time series of Picarro and TDLS traces for a continuous sampling from 5-8 May 2023. UTC time 300,000 corresponds to 11:19 local time on 5 May. (b) Scatterplot of 30-s averages of measurements from the TDLS (y-axis) and Picarro CRDS (x-axis).

Stability of the new TDLS was also assessed by examining three metrics of system performance, including detector signal at the start and end of each laser scan (representative of laser stability and optical efficiency), the ratio of these values (representative of laser and detector stability), and the position of the line center of the water vapor absorption feature (a direct measure of temperature of the laser TEC). In all experiments described here, the ratio of amplified detector signal at the start and end of each scan was found to vary by less than 2 % after subtracting the zero-signal measured when the laser is powered off. In addition, the center position of the water vapor line drifted by +/- 1 scan index point or less from scan to scan. Based on a calibration of the temperature dependence of line position using the setpoint of the PI controller to vary laser TEC temperature, it was found that this stability corresponds to < 0.001 K, a result that is consistent with the specifications of the WTC-3243.

## 4 Discussion

The goal of this work was to design, build, and characterize an economical and flexible fast-response instrument suitable for measurements of water vapor in the boundary layer. The entire electronics package is inexpensive and built with generalized components separated from the optical cell. A primary consideration was the use of low-cost, low power.





commercial off-the-shelf (COTS) components that, when combined with readily available lasers used by the telecom industry, allow for high-quality, high frequency observations at a fraction of the cost of commercial instruments with similar measurement characteristics. The key enabler for this new TDLS is the family of ARM-based microcontrollers based on the Cortex-M4 RISC integrated circuit. In this case, one controller is dedicated to controlling the laser in a highly reproducible manner required for maintaining tight temperature control with a commercial PI temperature controller package. In large part,

the use of highly efficient microcontrollers resulted in a system that consumed only 2 W and could run for several hours on a pair of small, rechargeable batteries. The resulting total hardware cost of the instrument is mainly due to the laser, detector, and optics. The remaining components (Teensy's, board and various electronics) total ~$300.

A list of components with manufacturer, model, mass, power consumption, and price at the time of purchase, is shown in Table 1:


| Component | Part # | Mfc. | Mass (g) | Cost ($) | Power (W) |
|---|---|---|---|---|---|
| Electronics Box | | | 417 | 25 | n/a |
| Custom printed circuit board | | OSH Park | 36 | 65 | - |
| Distributed Feedback Laser | NLK1E56AA | NTT Innovative Devices | | 1700 | 0.325 |
| Temperature controller | WTC 3243 | Wavelength Electronics | | 100 | 0.50 |
| Microcontrollers | Teensy (3.6 or 4.1) | PJRC | | 60 | 0.80 |
| Power conditioning | misc | misc | | 20 | 0.40 |
| Batteries | ARB-L16-700UP | Fenix | | 20 | ** |
| Detector amplifier circuit | misc | misc | | 15 | 0.025 |
| Collimating lens, card cage, mounts | | Thorlabs | 916 | 300 | n/a |
| InGaAs detector | FD1500 | Fermionics | | 200 | n/a |
| **Total** | | | **1333** | **2500** | **2.05** |

Since this project was undertaken, the Teensy family of microcontrollers was impacted by global supply chain shortages of chips. Thus, the Teensy 3.6 is no longer available, and an alternative is needed to drive the laser. The primary consideration is that the laser driving function must be highly reproducible, both in ramp frequency and in power, to maintain

precise tuning of the DFB output wavelength across the scan window. Replicating the measurements shown here would require generating ~1000 points per scan at a rate of 10 Hz (i.e., 10 ksps), with 12-bit resolution and uniform time steps for each update of the DAC. Several microcontrollers have demonstrated this level of performance, including the ItsyBitsy M4 Express, which also employs the Cortex-M4 processor and fast 12-bit true analog DAC. It would also be straightforward to use the Teensy 4.1



digital lines to drive a commercial DAC chip such as the AD5638 series from Analog Devices. Also noteworthy, we have carried out some tests showing that full scans over the water of ~1000 Hz are possible with the Teensy 3.6 and some of these alternatives, potentially enabling high-accuracy sampling at 10-to100-times the rates shown here, albeit with reduced precision.

Throughout the course of this work we experimented with other designs, including the components that convert the voltage into current to drive the laser output, different configurations for the transimpedance amplifier, and lower voltage electronics that allow for operation off a single 3.6 V lithium battery. In all cases similar high performance was maintained. For example, we have successfully tested powered the laser with a miniature low-power diode laser driver (FL500, Wavelength Electronics, Bozeman, MT). The FL500 also offers additional useful features such as overvoltage protection and enable/disable pins to protect the laser. Out of convenience, all the results shown here were obtained with a simple transimpedance amplifier circuit with the op-amp powered by 5 V, and with zero bias on the InGaAs detector. It is possible to further reduce detector/amplifier noise by biasing the InGaAs detector with -2.5 V. Finally, we have successfully demonstrated that significantly lower power consumption is possible by using components that operate at 3.3 V, thus eliminating the need for two 3.6 V batteries in series.

One of the initial goals of this work was to develop a package that allows for quick swapping of lasers and optics in the field. This is achieved by using a DFB laser in a standard butterfly package with integrated with TECs and fiber-coupled FC/APC connector. Such an approach allows for swapping electronics with different lasers for probing different gases or for swapping optical systems allowing for different optical path lengths required to achieve adequate sensitivity, including options for employing folded optics such as Herriott Cells or retroreflectors. Future applications envisioned by our laboratory including measurements of water vapor from stratospheric balloons, configuring for use on small unattended aerial vehicles, and autonomous measurements from meteorological stations in remote locations, such as on buoys, the Antarctic plateau, or mountain peaks.

## 5 Conclusion

We have developed an economical and flexible fast-response tunable diode laser spectrometer suitable for measurements of water vapor in the atmospheric boundary layer (ABL). The instrument bridges the current gap between research grade instruments costing tens of thousands of dollars and low-cost sensors commonly employed in portable meteorological stations and hand-held devices. The novel feature of the new TDLS is the use of a pair of low-cost, low power microprocessors based on the Cortex-M4 ARM family of integrated circuits. A series of intercomparisons with existing instruments used for high-accuracy measurements of water vapor, including for eddy covariance, demonstrates that the new TDLS is well suited for similar measurements for a fraction of the cost of existing instruments. Such a capability allows users with little previous expertise in instrumentation to acquire high quality, fast-response observations of water vapor for a variety of applications, including frequent horizontal and vertical profiling by uncrewed aerial vehicles, long-term eddy covariance measurements





from fixed and portable flux towers, and routine measurements of humidity from weather stations in remote locations such as the polar ice caps, mountains, and glaciers.

*Code availability:* The extraction codes and Arduino sketches are available open source on GitHub.

*Data availability:* The data used in this paper are available from the corresponding author upon request.

*Competing Interests:* Some authors are members of the editorial board of journal AMT.

*Author Contributions:* DT conceived and managed the project, including acquiring funding. The new TDLS was designed and fabricated by DT, EW, and LK. EW developed code for operating and extracting data from the TDLS. EW performed
experimental work and data analysis, with assistance from DT. Drafting of the manuscript was coordinated by EW with contributions from all three authors.

*Acknowledgements:* We thank David Noone and Adriana Bailey for assistance with operation and maintenance of the Picarro CRDS. We thank Scott Kittelman for access to the ATOC Skywatch Observatory and for technical support for field measurements.

*Financial support:* Seed funding for this project was provided by the University of Colorado. Some material is based upon work supported by the National Science Foundation under Grant No. AGS-2233136 and by the National Aeronautics and Space Administration Earth Sciences Division, Award No. 80NSSC20K0729. Any opinions, findings, and conclusions or recommendations expressed in this material are those of the authors and do not necessarily reflect the views of the National Science Foundation, NASA, or the University of Colorado.



.




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
