# Peer review of "An Economical Tunable-Diode Laser Spectrometer for Fast-Response Measurements of Water Vapor in the Atmospheric Boundary Layer"

_Atmospheric Measurement Techniques, 2024_

## Referee Comment (RC2)

Review of Wein et al., An Economical Tunable-Diode Laser Spectrometer for Fast-Response Measurement of Water Vapor in the Atmospheric Boundary Layer, AMT-2024-34, June 2024

The manuscript by Wein, et al. describes the recent development of a lightweight, low-cost, open-path TDLAS instrument suitable for research quality, fast-response measurements of water vapor in the atmospheric boundary layer and is highly appropriate for publication in AMT. The manuscript is well structured and describes details of the development and performance of the instrument that could reasonably allow reproduction of the instrument for use by others and includes discussion of a number of possible applications. The manuscript could use a number of minor corrections and modifications and I recommend publication following minor revision and considering of the following specific comments and questions.

Specific comments:

L8:     First sentence could be restructured to remove redundancy of "high spatiotemporal variability" and "abundances varying…"

L8:     ABL should be defined here rather than L11, and then L11 could just use "ABL"

L8:     "possesses" should be "poses" (or "presents")

L11:    "in situ" is a Latin term and not hyphenated. I would think that you would want to include "open-path" in the description here since that is a critical aspect enabling the fast response time.

L11:    "tunable diode" is generally not hyphenated (although I see it also was in Dorsi et al 2014).

L11:    You define 'TDLS' first in the abstract as "tunable diode laser spectroscopy", but then use it and subsequently define it as "tunable diode laser spectrometer". Perhaps use "tunable diode laser absorption spectroscopy" in the abstract since that is the technique utilized and then TDLS as the spectrometer.

L12: you only need to include the acronym definition here if you will use the acronym alone later in the abstract. Comment also applies to L16 and L21.

L16:    "proportional – integral", as appears in L104 of the text

L17:    "comprised of" should technically be "composed of" or "comprises"

L18:    perhaps "agreed"

L19:    perhaps "will allow" and preface with something like "The instrument is robust and simple to operate"

L29:   "tropics"

L32:   "underlying mesoscale processes"—meteorologically, mesoscale is typically 10 to 100s of km, which doesn't seem appropriate here

L39:   "DIALs and Raman lidars" or "differential absorption and Raman lidars"

L45: "such as infrared gas analyzers (IRGAs)"

L46:   "have come to" → "are typically used to"

L47:   it is really the cost (~x10, and for some applications the size/weight), not the limited number of vendors or some "highly specialized" nature that is the limitation, right? And potentially differences in required maintenance/recalibration? You expect the new TDLS to not require recalibration (does require initial calibration per L177) or regular maintenance, correct?

L57:   "prediction"

L57:   "capable of"

L60:   "development and performance"?

L61:   "high accuracy and precision matching that of" and "lower cost and greater flexibility that would allow widespread deployment for routine observations"

L67:   "laser diode"? and what is meant by a "generic" package since it does require built-in TEC and tight coupling of the fiber?

L69:   "components"? and "components and exhibits"

L71:   I'm a little skeptical of the emphasis on the use of the instrument by fully inexperienced operators

L76:   "2023), the reported instruments have had a slow response, resulting in limited vertical resolution"

L78:   an example of a location?

L81:   what is meant by "terrain and variable inhomogeneity"?

L89:   "based on" would be more appropriate

L90:   the clause "a schematic of which is shown in Fig. 1." currently references the previously reported (Dorsi et al 2014) instrument. The clause could be inserted immediately after "described here" in L89 to be clear

L92:   "is rapidly scanned"; "variations, a short"

L100:  Figure 1 shows the trigger pulse passing from the receiver microcontroller to the laser drive, but the text states that the trigger pulse for data collection originates from the laser driver board.

L104:  "TEC controller"

L105:  "temperature of 0.002K" should be "temperature of XX.XXX ± 0.002 K" or say "A temperature stability of ±0.002 K, consistent…"

L107:  "DFB" should be "laser" (or "DFB laser diode")

L108:  "a digital-to-analog (DAC) output" since the 3.6 has two, although the 4.1 does not have a DAC, so only from the 3.6 (although, as noted, now discontinued)

L110:  "Arduino-compatible" hyphenated? But not "laser driving" or "data acquisition"

L111:  "based on"

L112:  "previous instruments" developed in your lab? Or universally?

L117:  "scans to ~10 kHz and faster, resulting in high precision of the measurements"—precision from averaging over multiple scans? Current operation is only 10 Hz (100 msec) scans? L320 says "tests showing that full scans over the water [line] at ~1000 Hz are possible" and that higher scan (measurement) rates result in reduced precision (for individual scans)

L120:  Reference to Figure 2 is missing from the text (~L129?). Fig 3 is already mentioned on L126. Reorder sentences to put "Prior to…" after the circuit discussion? Would it make sense to include Figure 2 in supplemental material? That would allow additional inclusion of the custom TIA circuit and supporting circuit board.

L129:  "A Teensy model 4.1 with a built-in Micro-SD card feature was used…"; "a trigger pulse"

L131:  ADC not defined at first use; "data acquisition analog-to-digital conversion (ADC) is started."?

L132:  There is some discrepancy regarding the discussion of Fig 3. It says here that the plot contains 445 points, but the figure shows 4 complete scans. Figure 4 shows 445 points

without showing a complete scan (~10+425+~10?). It would be best to clearly describe the sequence of one scan (475 points? 30 + 425 + 20?) and show the complete scan in Figure 4.

L132: How does the math for 7.2 kHz "raw" ADC work with 475 pts / 100 msec at 32x oversampling? Does 7.2 kHz already include the 32x and so is faster than the 4750 samples / sec?

L144: omit "on the opposite side of optical path both operated in photovoltaic mode"? A following sentence begins "The photodiode is operated in photovoltaic mode"

L148: It seems like the "AD1101, Analog Devices" is actually "HMCAD1101"? I could not find a part at Analog Devices that was just "AD1101".

L165: description here is "1$^{st}$-order polynomial" while the caption in Fig 4 uses "linear fit"— these are indeed the same thing, but it might be clearer to be consistent

L166: It would be useful to have a little more clarity on the process of converting the temperature – wavelength determination to the current ramp scan to account "for the possible drift of the tune temperature by removing the nonlinear output laser wavelength response to a linear current ramp" and determination of the scan wavelength range.

L182: Accuracy metrics of the BMP280?

L187: Does "These calculations" refer to the real time processing that is planned for future implementation and not the present version that is the focus of the manuscript?

L190: The units of the x axis in Figure 4(b) are wavenumber, not wavelength as stated. Since wavelength is otherwise used consistently in the manuscript, I would suggest using wavelength here as well.

L190: In Figure 4(A), it might be helpful to use color on the trace to highlight the region of the scan used for the baseline fit. As noted in L132 comment, it would be clearer to plot a full scan including the 30 and 20 detector zero (laser off) points at the beginning and end of the scan.

L198: no hyphen needed between number and unit "25 L" even when used as an adjective

L200: "saturated to a mixing ratio of ~27,000 ppm"—was the air in the chamber saturated (potential condensation)? Or was the saturation temperature of the generator lower than the ambient temperature? What is/are the values (uncertainty) of the mixing ratio reported by the reference CRDS measurement rather than "~"?

L201:  "admitted to the chamber"—also flow out of the chamber as well to maintain P?

L207:  It would be good to include information about the linear regression as text in Figure 5

L209:  remove period following "points)"; omit reference

L213:  "Allan variance"

L222: sensitivity is not affected by averaging—"detection limit"?

L256:  Omit "A long electrical line"? This was replaced with the "10 m twisted pair cable"? No comma needed after "cell" or hyphens between numbers and units; the word "long" could be omitted

L307:  "Teensys"

L320:  "tested powered"

L336:  "include"

L342:  ABL already defined in introduction

---

## Author Comment (AC1)

**Author Comments to Referee Comments #1:**

We thank the referee for thoughtful comments that will improve the clarity of the paper. We address each of the comments below, with more detailed responses when appropriate. Reviewer comments are in black text with our responses in blue text.

The manuscript describes the promising development of a low-cost, fast-response tunable-diode laser spectrometer for measuring water vapor in the atmospheric boundary layer. As the instruments hard and the software design is based on readily available components, lightweight, and energy-efficient it can be easily reproduced and exhibits a broad spectrum of potential atmospheric measurement applications. The laboratory calibration and first atmospheric measurements show a high accuracy and precision of 10 ppm at 10Hz allowing for sampling with high spatial and temporal resolution.

Thank you. We are excited about the new instrument and feel that it is important to share sufficient details so that others who are interested may be able to build a similar one.

**General:**

- The consistency of naming should be improved throughout the manuscript. Some names are changed multiple times as an example: Teensy = Teensy 4.1 = data collection microcontroller = receiver. Also introduced acronyms should be used or left out. More consistency here would be very helpful.

These are noted and will be rectified throughout the manuscript.

- The quality of language/sentence syntax should be improved. Many typos, slips of the pen, and complicated/convoluted sentences.

Thank you. We will proofread the manuscript with an eye on improving the readability.

- The plotting ticks should be changed using to a more standard style as they are confusing that way. E.g.: Fig 5, yaxis highest tick should be 30, above the axis a common coefficient with $1 \times 10^{-3}$.

The figures will be modified to be more consistent with other recent publications in AMT.

- Also cross check references if missing and use correct citation style: Textual citation XXX et al. (YYYY) vs. parenthetical citations (XXX etl al. YYYY)

In-text references will be changed to address this issue and use textual citation when appropriate.

**2.1 Hardware Description**

- The captions of Fig.1 to Fig. 3 should be extended to be self-explanatory and not reference to the text.

For example: Fig 1, please explain the different components shown: are drive input and TEC input circuits by themselves, how does the TIA look, are the laser driver and data collection unit the two Teensy microcontrollers…

The figure captions will be expanded or modified to stand alone and not reference the main body of the paper. figures in the caption to explain more of what they depict.

- L 103: The laser *wavelength* is tuned via temperature?

This statement will be improved.

- L 105: add ± before 0.002 K

Thank you; ± will be added.

- L 127: adapt trigger pulse direction in Fig 1. From driver to data collection if the text is correct

This will be corrected in the figure.

- L129: After 5V reference to figure

This reference will be added and describes a custom circuit whose location is depicted in figure 1.

- L 141 – 146: Please clarify: Was the laser collimated or divergent? How much bigger is the laser beamwidth compared to the InGaAs sensor active area? Have there been any tests regarding vibrations? (Even collimated laser beams show a distinct Gaussian profile which could induce signal variations upon vibration)

Thank you for this important comment. We will clarify the approach. In fact, the collimating lens can be adjusted to expand the laser beam to more fully illuminate the detector. In this case, it would be divergent. We also performed tests with a collimated beam and found that the sensitivity to vibration was greater with more collimation. While this is not necessarily surprising, the main goal of this paper is to provide other practitioners with sufficient details of the electronics and control software so that they can develop their own systems for other applications. We used a simple lens/detector optical arrangement to prove that the method was both accurate and highly precise. It was mounted on a bracket that virtually eliminated vibration, and so the measurements did not experience noise due to vibration. A more flexible optical mount would, in fact, be more sensitive to vibration and we advise the experienced practitioner to design the optics with care. There are multiple examples of different optical designs in the literature, and we have added a few to the reference list.

- L 149: Syntax wrong or double sentence

This will be corrected.

**2.2 Spectral Processing**

- L 163: please cross check values on where the offset/drift is determined 30 & 20 points vs. 10 point in fig 4 caption.

This inconsistency will be addressed, and the figure caption will be corrected.

L 165 – 169: This part requires more clarification on how to convert scan steps into wavelength.

We agree. Although it is a straightforward process, it is relatively detailed. We will expand on this description.

- L 170: What do you want to say with the "[..] are then placed in an array?

This sentence will be deleted as this level of detail is unnecessary.

- Fig4 (b) Unit should be wavenumber

This will be corrected.

**3. Results:**

- Fig 5: Does the x-axis represent values measured by the Picarro or converted values from the TDLS?

  Please also state the lin-regress function parameters (slope, intercept, $r^{2)}$ or a plot of the converted H2O-ppm from TDLS over H2O-ppm measured by the Picarro together with a 1:1 line and respective regression slope. Similar as for Figure 8 (b)

Information for fit parameters will be added to figure 5 caption. X-axis represents values measured by Picarro.

- L 216 – 220: This part needs more clarification. Was a different InGaAs sensor used for the calibration of the instrument than the actual measurement? That could yield a different conversion coefficient.

Thank you for pointing this out. We used multiple detectors throughout the course of this work, and we will clarify our usage of detectors in this work.

L 239: Wu et al 2015 citation does not present in references

The citation to Wu et. al 2015 will be added.

- Fig8 (a). Please convert the x-axis to actual time in UTC

Time will be converted to UTC.

- L 278: Doesn't the averaging over 30s smooth all variation on a spatial scale of 1.5m? Hence I am not surprised that that values align even the locations had a 1.5m separation.

We agree, and we will add a short statement to this effect.

**Some Typos:**

- L 104: remove dot after "([…], MT)"
- L 111: add quotes " after receiver
- L 122: resister -> resistor
- L 209: remove dot after "(black points)"
- L 256: lowercase a before 25-m long
- L 298: Remove dot after power
- L 320: mission word(s) after water
- L 325: Either tested or powered
- L 333: remove with before TEC
- L 337 delete "configuring for use"
- L 342: ABL was already introduced in the introduction

Thank you for the careful review of the paper. These typos would be corrected.

---

## Author Comment (AC2)

**Author Comments to Referee Comments #2:**

We thank the referee for thoughtful comments that will improve the clarity of the paper. We address each of the comments below, with more detailed responses when appropriate. Reviewer comments are in black text with our responses in blue text.

Review of Wein et al., An Economical Tunable-Diode Laser Spectrometer for Fast-Response Measurement of Water Vapor in the Atmospheric Boundary Layer, AMT-2024-34, June 2024

The manuscript by Wein, et al. describes the recent development of a lightweight, low-cost, open-path TDLAS instrument suitable for research quality, fast-response measurements of water vapor in the atmospheric boundary layer and is highly appropriate for publication in AMT. The manuscript is well structured and describes details of the development and performance of the instrument that could reasonably allow reproduction of the instrument for use by others and includes discussion of several possible applications. The manuscript could use several minor corrections and modifications and I recommend publication following minor revision and considering of the following specific comments and questions.

Thank you. We hope that this paper will allow other investigators to develop similar instruments for their studies, and we are eager to see what future applications may be enabled by our work. We will make the suggested corrections.

Specific comments:

L8: First sentence could be restructured to remove redundancy of "high spatiotemporal variability" and "abundances varying…"

The redundancy will be removed.

L8: ABL should be defined here rather than L11, and then L11 could just use "ABL"

This will be corrected.

L8: "possesses" should be "poses" (or "presents")

We will make this change.

L11: "in situ" is a Latin term and not hyphenated. I would think that you would want to include "open-path" in the description here since that is a critical aspect enabling the fast response time.

Thank you. We will add "open-path" and correct hyphenations.

L11: "tunable diode" is generally not hyphenated (although I see it also was in Dorsi et al 2014).

We will search on, and correct hyphenations, throughout.

L11: You define 'TDLS' first in the abstract as "tunable diode laser spectroscopy", but then use it and subsequently define it as "tunable diode laser spectrometer". Perhaps use "tunable diode laser absorption spectroscopy" in the abstract since that is the technique utilized and then TDLS as the spectrometer.

This will be clarified.

L12: you only need to include the acronym definition here if you will use the acronym alone later in the abstract. Comment also applies to L16 and L21.

These will be clarified.

L16: "proportional – integral", as appears in L104 of the text

Thank you for noting this inconsistency. We will change to the appropriate term.

L17: "comprised of" should technically be "composed of" or "comprises"

Thank you. We will correct this.

L18: perhaps "agreed"

Past tense will be used here and in other places where it refers to specific tests, as opposed to general performance characteristics.

L19: perhaps "will allow" and preface with something like "The instrument is robust and simple to operate"

Thank you. We will make these changes.

L29: "tropics"

We will change to lower case.

L32: "underlying mesoscale processes"—meteorologically, mesoscale is typically 10 to 100s of km, which doesn't seem appropriate here

"Mesoscale" will be changed to "Microscale."

L39: "DIALs and Raman lidars" or "differential absorption and Raman lidars"

"DIAL and Raman lidars" will be changed to "differential absorption and Raman lidars."

L45: "such as infrared gas analyzers (IRGAs)"

This change will be made.

L46: "have come to" -> "are typically used to"

This change will be made.

L47: it is really the cost (~x10, and for some applications the size/weight), not the limited number of vendors or some "highly specialized" nature that is the limitation, right? And potentially differences in required maintenance/recalibration? You expect the new TDLS to not require recalibration (does require initial calibration per L177) or regular maintenance, correct?

Yes, thank you. There are several factors that limit the availability of relatively inexpensive instruments. We have decided to modify this point to make it clear that our instrument is not designed as a substitute for high-quality commercial instruments that have served the community well for many decades. Rather, we hope that our approach enables new applications that are limited by cost. In addition, it is our view that by providing the raw data necessary to calculate absorbances, the need for frequent calibrations is reduced. However, maintenance will depend upon the particular application (e.g., cleaning of optics to remove dust, etc.).

L57: "prediction"

This change will be made.

L57: "capable of"

This change will be made.

L60: "development and performance"?

This change will be made.

L61: "high accuracy and precision matching that of" and "lower cost and greater flexibility that would allow widespread deployment for routine observations"

This change will be made.

L67: "laser diode"? and what is meant by a "generic" package since it does require built-in TEC and tight coupling of the fiber?

 "Generic" will be removed and we will replace with "butterfly."

L69: "components"? and "components and exhibits"

This change will be made.

L71: I'm a little skeptical of the emphasis on the use of the instrument by fully inexperienced operators.

This is a fair statement. We will clarify this to mean operators without extensive experience with laser spectroscopy.

L76: "2023), the reported instruments have had a slow response, resulting in limited vertical resolution"

This change will be made.

L78: an example of a location?

We will provide examples of such locations.

L81: what is meant by "terrain and variable inhomogeneity"?

this will be changed to "heterogenous scalar and vector fields resulting from complex terrain."

L89: "based on" would be more appropriate

This change will be made.

L90: the clause "a schematic of which is shown in Fig. 1." currently references the previously reported (Dorsi et al 2014) instrument. The clause could be inserted immediately after "described here" in L89 to be clear.

This change will be made.

L92: "is rapidly scanned"; "variations, a short"

Noted; however this change conflicts with the intended meaning of the sentence.

L100: Figure 1 shows the trigger pulse passing from the receiver microcontroller to the laser drive, but the text states that the trigger pulse for data collection originates from the laser driver board.

This inconsistency will be corrected.

L104: "TEC controller"

This change will be made.

L105: "temperature of 0.002K" should be "temperature of XX.XXX ± 0.002 K" or say "A temperature stability of ±0.002 K, consistent…"

Wording will be changed to "Temperature is maintained to ± 0.002 K of the setpoint,".

L107: "DFB" should be "laser" (or "DFB laser diode")

"DFB" will be changed to "Laser."

L108: "a digital-to-analog (DAC) output" since the 3.6 has two, although the 4.1 does not have a DAC, so only from the 3.6 (although, as noted, now discontinued).

Wording will be changed to "the digital-to-analog (DAC) output from the Teensy 3.6."

L110: "Arduino-compatible" hyphenated? But not "laser driving" or "data acquisition"

This change will be made.

L111: "based on"

This change will be made.

L112: "previous instruments" developed in your lab? Or universally?

This will be clarified.

L117: "scans to ~10 kHz and faster, resulting in high precision of the measurements"—precision from averaging over multiple scans? Current operation is only 10 Hz (100 msec) scans? L320 says "tests showing that full scans over the water [line] at ~1000 Hz are possible" and that higher scan (measurement) rates result in reduced precision (for individual scans)

Thank you. We meant "high resolution," albeit with reduced point-to-point precision. We will clarify this in the text.

L120: Reference to Figure 2 is missing from the text (~L129?). Fig 3 is already mentioned on L126. Reorder sentences to put "Prior to…" after the circuit discussion? Would it make sense to include Figure 2 in supplemental material? That would allow additional inclusion of the custom TIA circuit and supporting circuit board.

Reference to Figure 2 will be added. Because we do not have more substantial material for a supplement, we will come up with a way to provide adequate information to laser driving and detector amplifying circuits. In fact, both were quite simple and adapted from information that is readily available on the World Wide Web. We decided not to provide the actual circuit diagrams or fabrication files because they are fairly limited to the specific components we used, one of which is no longer available. We will add a note that we can provide details of our electronics and circuit board fabrication upon request.

L129: "A Teensy model 4.1 with a built-in Micro-SD card feature was used…"; "a trigger pulse"

This change will be made.

L131: ADC not defined at first use; "data acquisition analog-to-digital conversion (ADC) is started."?

This change will be made.

L132: There is some discrepancy regarding the discussion of Fig 3. It says here that the plot contains 445 points, but the figure shows 4 complete scans. Figure 4 shows 445 points without showing a complete scan (~10+425+~10?). It would be best to clearly describe the sequence of one scan (475 points? 30 + 425 + 20?) and show the complete scan in Figure 4.

Fig. 3 shows the continuous signal output from the TIA circuit as recorded by an oscilloscope. Fig. 4 shows a complete scan read from the TIA circuit by the Teensy 4.1. There is dead time between scans not shown in Figure 4 for when data are written to the microSD card. We will clarify this in the text.

L132: How does the math for 7.2 kHz "raw" ADC work with 475 pts / 100 msec at 32x oversampling? Does 7.2 kHz already include the 32x and so is faster than the 4750 samples / sec?

The Teensy 4.1 samples at 300,000 samples per second. Therefore, the full bandwidth is 9375 samples per second with 32x oversampling. At 100 ms we use 50% of this available bandwidth in order to slow down the "write-to-SD" process and provide overhead for performing calculations, as desired. We will clarify this in the text.

L144: omit "on the opposite side of optical path both operated in photovoltaic mode"? A following sentence begins "The photodiode is operated in photovoltaic mode"

This change will be made.

L148: It seems like the "AD1101, Analog Devices" is actually "HMCAD1101"? I could not find a part at Analog Devices that was just "AD1101".

Correct part number is LT1013 CN8. This will be changed in the text.

L165: description here is "1st-order polynomial" while the caption in Fig 4 uses "linear fit"— these are indeed the same thing, but it might be clearer to be consistent.

 "1st-order polynomial" will be replaced with "linear fit" where applicable.

L166: It would be useful to have a little more clarity on the process of converting the temperature – wavelength determination to the current ramp scan to account "for the

possible drift of the tune temperature by removing the nonlinear output laser wavelength response to a linear current ramp" and determination of the scan wavelength range.

We agree. Although it is a straightforward process, with a result that is fairly unremarkable (e.g., a linear conversion from bit number to wavelength), it requires a fairly lengthy discussion. However, we have decided to add a short explanation of how we measured the conversion, and we will provide the specific equation for our specific scan parameters. In addition, we will note that the conversion depends on specific conditions of the ramp.

L182: Accuracy metrics of the BMP280?

The details of the BMP280 will be added.

L187: Does "These calculations" refer to the real time processing that is planned for future implementation and not the present version that is the focus of the manuscript?

Yes. We are in the process of developing embedded codes for processing spectra in real time for future versions of the instrument. This clarification will be made.

L190: The units of the x axis in Figure 4(b) are wavenumber, not wavelength as stated. Since wavelength is otherwise used consistently in the manuscript; I would suggest using wavelength here as well.

Units will be adjusted.

L190: In Figure 4(A), it might be helpful to use color on the trace to highlight the region of the scan used for the baseline fit. As noted in L132 comment, it would be clearer to plot a full scan including the 30 and 20 detector zero (laser off) points at the beginning and end of the scan.

These highlights will be added.

L198: no hyphen needed between number and unit "25 L" even when used as an adjective.

This change will be made.

L200: "saturated to a mixing ratio of ~27,000 ppm"—was the air in the chamber saturated (potential condensation)? Or was the saturation temperature of the generator lower than the ambient temperature? What is/are the values (uncertainty) of the mixing ratio reported by the reference CRDS measurement rather than "~"?

The sentence starting on L200 will be changed to "A beaker of warm water was placed inside the chamber to humidify the air to 29,000 ppm, just below the saturation point. Over the course of the next two hours water vapor mixing ratios in the chamber were reduced to ~13,000 ppm by stepwise-addition of relative dry ambient air from the laboratory."

L201: "admitted to the chamber"—also flow out of the chamber as well to maintain P?

This will be clarified.

L207: It would be good to include information about the linear regression as text in Figure 5

The fit parameters will be included in the Figure 5 caption.

L209: remove period following "points)"; omit reference

This change will be made.

L213: "Allan variance"

This change will be made.

L222: sensitivity is not affected by averaging— "detection limit"?

Thank you for noting this ambiguity. We will clarify when we mean precision and/or limits of detection.

L256: Omit "A long electrical line"? This was replaced with the "10 m twisted pair cable"? No comma needed after "cell" or hyphens between numbers and units; the word "long" could be omitted.

This change will be made.

L307: "Teensys"

This change will be made.

L320: "tested powered"

This change will be made.

L336: "include"

This change will be made.

L342: ABL already defined in introduction

This change will be made.

---

## Author Comment (AC4)

Response to Editor, https://doi.org/10.5194/amt-2024-34-EC2

(Editor Comments in black, responses in red)

The authors submitted a substantially revised version on the basis of the reviewers' comments.

However, after reading the revised version, I find that quite a number of points still do need improvement, for instance:

> We are grateful for the Editor's careful scrutiny of the revised manuscript and track-changes version. While there were many revisions, the majority were relatively minor. The program we used to track changes created an unnecessarily long document. It has been significantly shortened by removing tracking for small changes such as punctuation, ordering and position of references in sentences, and formatting. We have added several more substantive changes as recommended by the Editor, which we summarize below. We are pleased to make these changes as they improve the paper.

The sentence in lines 58-61 introducing the new instrument is very hard to read (in fact the sentences in the original version of the manuscript were better in that respect):

"As fast in situ observations of $H_2O$ are essential for numerical weather prediction and for investigations of the evolution of the ABL and its turbulence characteristics (e.g. large eddy simulations), and there is a need for more frequent measurements from remote locations, we have developed an economical new fast-response laser spectrometer (Helbig et al., 2021; Petersen, 2016)."

> We have changed this to read:

> "High-resolution in situ observations of $H_2O$ are essential for numerical weather prediction and for investigations of the evolution of the ABL and its turbulence characteristics (e.g. large eddy simulations), and there is a need for more frequent measurements from remote locations (Helbig et al., 2021; Petersen, 2016)."

Line 65: The term „capacitive sensors" appears to correspond to "thin-film water-sensitive polymers sandwiched between two electrodes" in lines 53-54. This should be made clear.

We agree. We have added changed this to read:

> "At the other end of the cost spectrum are various versions of capacitive humidity sensors that employ thin-film water-sensitive polymers sandwiched between two electrodes."

Lines 83-84: The meaning of „heterogeneous scalar and vector fields resulting from complex terrain" remains cryptic. If H2O mixing ratio (scalar) fields and H2O flux (vector)

> We have modified the sentence to read:

"Another application is tracking water-resource loss from reservoirs with ground-based flux measurements. There is a need to increase the density of measurements on specific reservoirs to map out the large spatial and temporal gradients in humidity due to adjacent complex terrain that contributes to significant errors in latent heat fluxes derived from those measurements (Friedrich et al., 2018)."

Line 105 (Caption of Fig. 1): What does the dashed line (there appears to be no dotted line) actually indicate ?

Thank you for catching this error. We changed line styles for clarity without correcting the figure caption. This, we have modified the figure caption to read:

"The components surrounded by the bold dashed line are contained on a single printed circuit board (schematic shown in Fig. 3)."

Line 125: Explain "a Teensy 3.6". Is this one of the Arduino micocontrollers mentioned above?

Correct. For clarity, we have modified this to now read:

"Two independent Arduino-compatible microcontrollers (PJRC, Sherwood, OR) were chosen for separately driving the laser (a Teensy 3.6) and for data acquisition (a Teensy 4.1)."

In addition, we have added "3.6" and "4.1" to the labels in Figure 1 to clearly indicate the locations of these two microcontrollers.

Line 141: What does the term „Teensy 4.1" refer to?

The Teensy 4.1 is the microcontroller that acquires and stores data. Hopefully this is now clarified by the change above.

Line 179: Give figures also in nm.

We have converted from wavenumbers to nm: "1373.3002 and 1373.2878 nm"

Figure 3 (new): Lower right corner: TIP32AG is designated PNP transistor, but the symbol indicates an NPN transistor (which is probably correct from the schematics, so transistor type designation is probably wrong), perhaps TIP31AG.

Thank you for catching this error in the PNP transistor symbol in the schematic. This has been corrected.

Unfortunately this list can be extended.

Thank you. Without specific examples, we have carefully reviewed the remainder of the manuscript and made a few additional changes, which we have noted in the new versions of track changes and authors' response. We trust that these will adequately address the Editor's concerns, but we are open to further editorial or copy-edits that improve the final manuscript.

Furthermore the „track changes" document is not very helpful in that several sections of the text are marked as changed while they are actually unchanged (e.g. most of lines 58 and 59). At the same time several changes (e.g. the caption of the new Fig. 3) are not marked as changed although they are clearly new text.

The Editor has raised a good point; the track-changes version that was generated by our editing software was verbose. We have taken another pass through the marked-up version for the Editor, and we have manually removed instances where the original document is unchanged.

Also the line numbers given in the answer to the reviewers do not appear to match the text.

Thank you. Unfortunately, we do not understand what happened with the document received by the Editor, as it does not match the line numbering on the version we submitted. We do notice that a marked-up document generated by an iMac have different line numbering that one generated by a Windows machine. However, that does not completely explain the discrepancy. We will work with the Editorial Support Team to ensure that the Editor receives a copy that has numbering that corresponds to the version we have used for the "explanation of changes" document.

Overall, I am sorry to have to conclude that the revised manuscript is still not ready for acceptance for final publication in AMT.

I therefore suggest:

1) That the authors provide a further revise version after thoroughly going through the text and taking care of the unclear statements and errors. In this context it might be helpful if a more senior scientist (co-author) could help.

As the most senior scientist on the team, Prof. Toohey has created the versions of the revised documents submitted to Copernicus for final review. We welcome any suggestions, including at the copy edit stage, that will further improve the final published version of the paper.

2) That the authors provide a „track changes" version that shows all changes (and only the changes) with correct line numbers.

This will be done.

Alternatively, the authors may re-submit a revised version of the manuscript to AMT.

This does not seem necessary, given the nature of the changes. We respect the view of the Editor and we accept this constructive recommendation and the spirit in which it was offered.

---

## Author Response (AR1)

**Justification of Changes**

**Per: Referee #1**

**General:**

- The consistency of naming should be improved throughout the manuscript. Some names are changed multiple times as an example: Teensy =   Teensy 4.1 = data collection microcontroller = receiver. Also introduced acronyms should be used or left out. More consistency here would be very helpful.

We have corrected the use of acronyms throughout. "Atmopsheric Boundary Layer" has been defined once and used throughout the manuscript. "Tunable diode laser spectrometer" has been defined once and used throughout as the name of the developed instrument. Labeling of Teensy microcontrollers have been cleaned up to limit references of microcontrollers to part name only.

- The quality of language/sentence syntax should be improved. Many typos, slips of the pen, and complicated/convoluted sentences.

We have corrected a number of typos and some sentences to improve the quality of the paper. They appear in the marked up version of the revised manuscript and will not be listed in detail here unless the changes were significant. We will summarize those significant changes at the end of this document.

- The plotting ticks should be changed using to a more standard style as they are confusing that way. E.g.: Fig 5, yaxis highest tick should be 30, above the axis a common coefficient with $1 \times 10^{-3}$.

Figure 5 y-axis changed.

Figure 6 y-axis changed.

Figure 8 y-axis a and b and x-axis b changed.

- Also cross check references if missing and use correct citation style: Textual citation XXX et al. (YYYY) vs. parenthetical citations (XXX etl al. YYYY)

Old L41: "(Wulfmyer et. al 2015)" moved to the end of the sentence. (New L40)

Old L58: "(Helbig et al., 2021; Petersen, 2016)" moved to end of sentence. (New L58)

Old L65: References changed to conform with AMT guidelines. (New L65)

Old L75: References changed here to conform with AMT guidelines for intext citations. (New L75)

Old L114: "(Rainwater 2022)" moved to end of sentence. (New L119).

Old L241: "(e.g. see Aslan et. al)" changed to "(Aslan et. al)". (New L262)

**2.1 Hardware Description**

- The captions of Fig.1 to Fig. 3 should be extended to be self-explanatory and not reference to the text.

  For example: Fig 1, please explain the different components shown: are drive input and TEC input circuits by themselves, how does the TIA look, are the laser driver and data collection unit the two Teensy microcontrollers…

Figure 1 caption (L100-104 in new document, or "new L100-104) has been expanded to include the following:

> "individual components (microcontrollers, laser, temperature controller) or individual circuits (TIA, laser driver). The dotted line indicates all components contained on the printed circuit board and those housed outside. A fiber optic coupler and twisted wire pair are passed outside electronics box through hermetically sealed holes."

Figure 2 caption (new L135-139) has been expanded to include the following:

> "Important components of the TDLAS laser scans as a function of time. Detector output (top panel) is the continuous voltage from the TIA. About one-third of the time the laser is off, and the signal is close to zero, the background for the detector and TIA circuit. Laser drive (middle panel) represents the voltage output by the Teensy 3.6 used to set the current of the laser. The trigger pulse signal (bottom panel) is sent by the Teensy 3.6 to the Teensy 4.1 to initiate the sampling and recording of the scan."

- L 103: The laser *wavelength* is tuned via temperature?

Old L103 has been changed to "…the laser is tuned to the wavelength of a strong water absorption feature at 1368.59 nm by changing the temperature of the laser diode…" (new L105)

- L 105: add ± before 0.002 K

Old L 105: "±" has been added (new L108)

- L 127: adapt trigger pulse direction in Fig 1. From driver to data collection if the text is correct

Figure 1 (new L100) has been correction to be consistent with the description in the text (new L128).

- L129: After 5V reference to figure

The old Figure 2 has been replaced with a new figure, now numbered Figure 3, which is a circuit diagram of the complete electronics used in the instrument. The following sentence has been added at new L127:

"A complete electronic circuit diagram of the instrument is shown in Fig. 3."

5V is now referenced on New Line 156 and New Line 165

- L 141 – 146: Please clarify: Was the laser collimated or divergent? How much bigger is the laser beamwidth compared to the InGaAs sensor active area? Have there been any tests regarding vibrations? (Even collimated laser beams show a distinct Gaussian profile which could induce signal variations upon vibration)

Old Lines 141-146 have been changed to (New L143-147):

"The lens was configured so that the laser beam was divergent to fully illuminate the active area of a low-noise broadband indium gallium arsenide (InGaAs) semiconductor photodiode and reduce variations in intensity due to vibration and turbulent fluctuations of air density in the optical path. Multiple photodiodes of differing manufactures (Thorlabs FDGA05, ThorLabs FGA04, Fermionics FD1500) were used over the course of this work, with no significant difference in results or performance." Now appearing in New L143-147.

- L 149: Syntax wrong or double sentence

Duplicate wording in L149 has been removed. The sentence now reads (New Line: 151-152).

"The gain was tuned using a 1-10 kΩ variable resistor."

**2.2 Spectral Processing**

- L 163: please cross check values on where the offset/drift is determined 30 & 20 points vs. 10 point in fig 4 caption.

Old L163 has been changed to (New L170):

"Briefly, a small detector/amplifier offset is determined from 10 points each at the start and end of each scan while the laser is powered off.".

The following sentence, which was in the original Figure 4 caption has been deleted (New L210).

"The initial ~10 points and final 10~ points represent the signal with the laser powered off."

The following sentence has been added to Figure 4 caption (New L211):

"The fit is made between the points highlighted in red (30 points at the start of the scan and 20 at the end)."

L 165 – 169: This part requires more clarification on how to convert scan steps into wavelength.

The following paragraph was added to address this (New L174-189).

"To account for possible drift of laser wavelength (e.g., the position of the absorption feature in a scan), a relationship between scan position and laser wavelength was estimated using a closely spaced pair of weak water absorption lines at 7281.72 and 7281.80 cm$^{-1}$ produced by a DFB laser-centered on a different wavelength than the one used for the measurements in this paper. The position of this pair was systematically scanned across the full temperature range of a single current ramp by slowly varying the setpoint of the WTC and the spacing between the two lines (0.08 cm$^{-1}$, or 0.015 nm) was measured in scan index (e.g., see Fig. 4). A linear fit to the ratio of this spacing to the difference in scan index as a function of scan position was determined as:

$s(x) \, (nm/step) = 0.00052 + x * 5.00 * 10^{-7}$

where s is the change in wavelength per scan index (of the 445 points) and x is the scan index. The use of this function results in a near-constant line width as a function of wavelength if the position of the absorption feature shifts slightly due to variations in laser baseplate temperature. Although such a shift was never observed in these experiments, it is a consideration for use in an environment where the ambient temperature may vary significantly – e.g., by many tens of degrees. This method also allowed for the determination of the full width of the scan to be 0.279 nm for the specific scan start and end points and scan rate used in these experiments."

- L 170: What do you want to say with the "[..] are then placed in an array?

We decided that this level of detail is unnecessary and so the following was deleted:

"The observed signal (i.e., Iobs(t)) and calculated background I0(t) are then placed in an array [i, Io(i), I(i)]."

- Fig4 (b) Unit should be wavenumber

Units in Fig. 4b have been changed to cm$^{-1}$ (i.e., wavelength).

**3. Results:**

- Fig 5: Does the x-axis represent values measured by the Picarro or converted values from the TDLS?

  Please also state the lin-regress function parameters (slope, intercept, $^{r2)}$ or a plot of the

converted H2O-ppm from TDLS over H2O-ppm measured by the Picarro together with a 1:1 line and respective regression slope. Similar as for Figure 8 (b)

Fig. 5 x-axis label changed from "$H_2O$ mixing ratio (ppm)" to "Picarro $H_2O$ mixing ratio (ppm)" and the following has been added to the caption of Figure 5 caption (New L230):

"Fit Parameters: slope = 0.0006, intercept = 0.0039, $R^2$ = 0.9999." (New L234)

- L 216 – 220: This part needs more clarification. Was a different InGaAs sensor used for the calibration of the instrument than the actual measurement? That could yield a different conversion coefficient.

The following has been added earlier in the manuscript (New L146):

"Multiple photodiodes of differing manufactures (Thorlabs FDGA05, ThorLabs FGA04, Fermionics FD1500) were used over the course of this work, with no significant difference in results or performance."

L 239: Wu et al 2015 citation does not present in references

Wu et al. 2015 has been added to the references.

- Fig8 (a). Please convert the x-axis to actual time in UTC

Figure 8 x-axis has been converted to UTC.

- L 278: Doesn't the averaging over 30s smooth all variation on a spatial scale of 1.5m? Hence I am not surprised that that values align even the locations had a 1.5m separation.

Old L281: The following has been added (New L301):

"…including variability due to the Picarro inlet and optical cell being separated by 1.5 m." (New L300).

**Some Typos:**

- L 104: remove dot after "([…], MT)"

Old L104: Period removed. (New L107)

- L 111: add quotes " after receiver

Old L111: " added after receiver. (New L114)

- L 122: resister -> resistor

Old L122: Spelling corrected. (New L110)

- L 209: remove dot after "(black points)"

Old L209: period removed. (New 231)

- L 256: lowercase a before 25-m long

Old L256: "A" changed to "a" (New L277)

- L 298: Remove dot after power

Old L298: period deleted. (New L320).

- L 320: mission word(s) after water

Old L320: "over the water" deleted. (New L341)

- L 325: Either tested or powered

Old L325: "successfully tested powered" changed to "successfully powered" (New L349)

- L 333: remove with before TEC

Old L333: "with" deleted. (New L353)

- L 337 delete "configuring for use"

Old L337: "Configuring for use" deleted. (New L357)

- L 342: ABL was already introduced in the introduction

Old L342 "atmospheric boundary layer (ABL)" changed to "ABL". (New L360)

**Per Referee #2:**

Specific comments:

L8: First sentence could be restructured to remove redundancy of "high spatiotemporal variability" and "abundances varying…"

Old L8: "The high spatial temporal variability of" deleted. Sentence now reads:

"Water vapor in the atmospheric boundary layer poses a significant measurement challenge with abundances varying by an order of magnitude over short spatial and temporal scales." (New L8-9)

L8: ABL should be defined here rather than L11, and then L11 could just use "ABL"

Old L11: "atmospheric boundary layer" deleted. "Atmospheric boundary layer" and abbreviation "ABL" are now introduced on New L25. Subsequent uses of "atmospheric boundary layer have been changed to ABL.

L8: "possesses" should be "poses" (or "presents")

Old L8: "possesses" changed to "poses". (New L8)

L11: "in situ" is a Latin term and not hyphenated. I would think that you would want to include "open-path" in the description here since that is a critical aspect enabling the fast response time.

Old L11: hyphen deleted in "in-situ". (New L10)

L11: "tunable diode" is generally not hyphenated (although I see it also was in Dorsi et al 2014).

Old L11: "Hyphen deleted" (New L10)

L11: You define 'TDLS' first in the abstract as "tunable diode laser spectroscopy", but then use it and subsequently define it as "tunable diode laser spectrometer". Perhaps use "tunable diode laser absorption spectroscopy" in the abstract since that is the technique utilized and then TDLS as the spectrometer.

The single instance where "TDLS" was used to refer to the method has been changed to "TDLAS" (New L10), and we retain the references to the instrument as the "TDLS."

L12: you only need to include the acronym definition here if you will use the acronym alone later in the abstract. Comment also applies to L16 and L21.

Old L12: "(SWIR)" is deleted as it is not used in other locations in text. (New L12)

L16: "proportional – integral", as appears in L104 of the text

Old L16: "proportional-integrating" changed to "proportional-integral" (New L16)

L17: "comprised of" should technically be "composed of" or "comprises"

Old L17: "comprised of" changed to "constructed of" (New L17)

L18: perhaps "agreed"

Old L18: "agrees" changed to "agreed" (New L18)

L19: perhaps "will allow" and preface with something like "The instrument is robust and simple to operate"

Old L19: "allows" changed to "is robust and simple to use and will" (New L19)

L29: "tropics"

Old L29: "Tropics" changed to "tropics" (New L29)

L32: "underlying mesoscale processes"—meteorologically, mesoscale is typically 10 to 100s of km, which doesn't seem appropriate here

Old L32: "mesoscale" changed to "microscale" (New L32)

L39: "DIALs and Raman lidars" or "differential absorption and Raman lidars"

Old L39: "DIAL and Raman lidars" changed to "differential absorption Lidars and Raman lidars." (New L39)
.

L45: "such as infrared gas analyzers (IRGAs)"

Old L45: "such as the infrared gas analyzers" changed to "such as infrared gas analyzers" (New L45)

L46: "have come to" -> "are typically used to"

Old L46: changed to "are typically used to" (New L45)

L47: it is really the cost (~x10, and for some applications the size/weight), not the limited number of vendors or some "highly specialized" nature that is the limitation, right? And potentially differences in required maintenance/recalibration? You expect the new TDLS to not require recalibration (does require initial calibration per L177) or regular maintenance, correct?

Text has been added or edited to address this points:

Old L47: Ending sentences changed to: "These research-grade instruments, which are used predominantly at multi-instrumented flux towers and weather stations, tend to be expensive, often costing $20,000 or more. In addition, they can incur additional costs for factory service to maintain high accuracy. Consequently, their use in remote locations has been relatively limited." (New L46-48)

Old L50-56: Paragraph edited to read: As fast in situ observations of $H_2O$ are essential for numerical weather prediction and for investigations of the evolution of the ABL and its turbulence characteristics (e.g. large eddy simulations), and there is a need for more frequent measurements from remote locations, we have developed an economical new fast-response laser spectrometer (Helbig et al., 2021; Petersen, 2016). The instrument is capable of fast measurements of water vapor in the ABL, while demonstrating high accuracy and precision comparable to that of commercially available research-grade commercial instruments. Built from low-cost components that are readily available commercially, the instrument exhibits relatively low up-front costs with the ability to replace critical components, thus bridging the gap between the more expensive and highly accurate fast-response instruments and the relatively inexpensive, but slower response capacitive sensors. (New L56-L63)

L57: "prediction"

Old L57: "predictions" changed to "prediction" (New L56)

L57: "capable of"

Old L59: "for" changed to "capable of" (New L59)

L60: "development and performance"?

Old L60:  Text edited as above (New L60)

L61: "high accuracy and precision matching that of" and "lower cost and greater flexibility that would allow widespread deployment for routine observations"

Old L61: "High accuracy/precision like" has been changed to "high accuracy and precision matching that of commercially available research-grade commercial instruments" (New L58).

Old L62: "low cost and flexibility desired for more" has been changed to "lower cost and greater flexibility" (New L60)

L67: "laser diode"? and what is meant by a "generic" package since it does require built-in TEC and tight coupling of the fiber?

Old L67: "generic" replaced with "common butterfly" (New L7)

L69: "components"? and "components and exhibits"

Old L69: "technology" has been changed to "components" (New L69)

L71: I'm a little skeptical of the emphasis on the use of the instrument by fully inexperienced operators.

Old L71: "research grade instruments" changed to "laser spectroscopy" (New L71)

L76: "2023), the reported instruments have had a slow response, resulting in limited vertical resolution"

Old L76: "the available instrumentation have slow response and limited vertical resolution" has been changed to "the instruments used have slow response times, resulting in limited vertical resolution" (New L76)

L78: an example of a location?

Old L78: We have added "remote land and ocean regions" and include a new citation to Brotzge, J. A., Berchoff, D., Carlis, D. L., Carr, F. H., Carr, R. H., Gerth, J. J., Gross, B. D., Hamill, T. M., Haupt, S. E., Jacobs, N., McGovern, A., Stensrud, D. J., Szatkowski, G., Szunyogh, I., and Wang, X.: Challenges and Opportunities in Numerical Weather Prediction, Bulletin of the American Meteorological Society, 104, E698–E705, https://doi.org/10.1175/BAMS-D-22-0172.1, 2023. (New L79)

L81: what is meant by "terrain and variable inhomogeneity"?

Old L81: terrain and variable inhomogeneity" changed to "heterogenous scalar and vector fields resulting from complex terrain" (New L81)

L89: "based on" would be more appropriate

Old L89: "off" changed to "on" (New L89)

L90: the clause "a schematic of which is shown in Fig. 1." currently references the previously reported (Dorsi et al 2014) instrument. The clause could be inserted immediately after "described here" in L89 to be clear.

Old L90: We have deleted "a schematic of which is shown in figure 1" and on Old L97 we have added "An overview of the instrument is depicted in Fig. 1." (New L96)

L92: "is rapidly scanned"; "variations, a short"

Old L91: (NLK1E56AA, NTT Innovative Devices, Yokohama, Japan) has been moved to New L90 to improve clarity.

L100: Figure 1 shows the trigger pulse passing from the receiver microcontroller to the laser drive, but the text states that the trigger pulse for data collection originates from the laser driver board.

Old L100: Figure 1 has been corrected to be consistent with the trigger signal described in the text.  (New L100)

L104: "TEC controller"

Old L104: "proportional-integral (PI)" changed to "PI TEC". (New L106)

L105: "temperature of 0.002K" should be "temperature of XX.XXX ± 0.002 K" or say "A temperature stability of ±0.002 K, consistent…"

Old L105: "±" has been added (new L108)

L107: "DFB" should be "laser" (or "DFB laser diode")

Old L107: "DFB" has been changed to "Laser" (New L111)

L108: "a digital-to-analog (DAC) output" since the 3.6 has two, although the 4.1 does not have a DAC, so only from the 3.6 (although, as noted, now discontinued).

Old L108: "output from one of the microcontrollers" has been changed to "digital-to-analog (DAC) output" (New L111)

L110: "Arduino-compatible" hyphenated? But not "laser driving" or "data acquisition"

Old L110: Hyphens deleted from "laser-driving" and "data-acquisition" (New L113)

L111: "based on"

Old L111: "off" changed to "on" (New L114)

L112: "previous instruments" developed in your lab? Or universally?

Old L112: We have added "developed in our lab and elsewhere, employing the same measurement technique as reported here" (New L115)

L117: "scans to ~10 kHz and faster, resulting in high precision of the measurements"—precision from averaging over multiple scans? Current operation is only 10 Hz (100 msec) scans? L320 says "tests showing that full scans over the water [line] at ~1000 Hz are possible" and that higher scan (measurement) rates result in reduced precision (for individual scans)

Old LM117: "10 kHz" has been changed to "1 kHz" (to reflect what has been implemented in lab) (New L120)

Old LM117: "and faster" has been changed to "being possible," and "precision" has been changed to "resolution" (New L120)

L120: Reference to Figure 2 is missing from the text (~L129?). Fig 3 is already mentioned on L126. Reorder sentences to put "Prior to…" after the circuit discussion? Would it make

sense to include Figure 2 in supplemental material? That would allow additional inclusion of the custom TIA circuit and supporting circuit board.

Based on this and the Editor's comments reiterating the request, we now include a circuit diagram of the entire instrument.

The old Figure 2 has been replaced with a new figure, now numbered Fig. 3, which is a circuit diagram of the complete electronics used in the instrument. The following sentence has been added at new L127:

"A complete electronic circuit diagram of the instrument is shown in Fig. 3."

Old L126: We have added (new L125):

"This voltage drives an operational amplifier (Analog Devices LT1101) that controls the current required to scan the laser from transistor (TIP 32AG n-channel JFET) in a textbook voltage-to-current converter circuit (Figure 6.31 of Horowitz and Hill, 1983)."

Old L148: We have added "The top panel in Fig. 2 shows the continuous output of this circuit." (New L150).

L129: "A Teensy model 4.1 with a built-in Micro-SD card feature was used…"; "a trigger pulse"

Old L130: "Upon receiving the trigger pulse, the internal clock is recorded into a buffer" has been changed to "Before the start of each scan, the Teensy 3.6 produces a digital pulse ("trigger"), shown on the bottom panel of Fig. 2, that initiates the data acquisition and storage process on a Teensy 4.1". (New L129)

L131: ADC not defined at first use; "data acquisition analog-to-digital conversion (ADC) is started."?

Old L131: Sentence now reads: Before the start of each scan, the Teensy 3.6 produces a digital pulse ("trigger"), shown on the bottom panel of Fig. 2, that initiates the data acquisition and storage process on a Teensy 4.1.  (New L129)

L132: There is some discrepancy regarding the discussion of Fig 3. It says here that the plot contains 445 points, but the figure shows 4 complete scans. Figure 4 shows 445 points without showing a complete scan (~10+425+~10?). It would be best to clearly describe the sequence of one scan (475 points? 30 + 425 + 20?) and show the complete scan in Figure 4.

Old L131: We have changed this to read:

"At this time, the internal clock is recorded into a buffer and the output from the detector TIA is recorded as a single scan consisting of 445 discreet sampled at 12-bit resolution.

Although the Teensy 4.1 samples at 300,000 samples per second, we oversampled 32 times using a software function that reduces noise inherent in the ADC." (New L130)

L132: How does the math for 7.2 kHz "raw" ADC work with 475 pts / 100 msec at 32x oversampling? Does 7.2 kHz already include the 32x and so is faster than the 4750 samples / sec?

Old L132: The change to Old L131 addresses this comment.

L144: omit "on the opposite side of optical path both operated in photovoltaic mode"? A following sentence begins "The photodiode is operated in photovoltaic mode"

Old L144: The phrase "(either Thorlabs FDGA05 or Fermionics FD1500) on the opposite side of optical path both operated in photovoltaic mode" has been deleted to address the comment from Referee 1. (New L148)

L148: It seems like the "AD1101, Analog Devices" is actually "HMCAD1101"? I could not find a part at Analog Devices that was just "AD1101".

Old L148: "AD1101" changed to "LT1013 CN8" (New L149)

L165: description here is "1st-order polynomial" while the caption in Fig 4 uses "linear fit"— these are indeed the same thing, but it might be clearer to be consistent.

Old L165: "1$^{st}$-order polynomial" changed to "linear fit" (New L173)

L166: It would be useful to have a little more clarity on the process of converting the temperature – wavelength determination to the current ramp scan to account "for the possible drift of the tune temperature by removing the nonlinear output laser wavelength response to a linear current ramp" and determination of the scan wavelength range.

Old L165: The following was added:

The following paragraph was added to address this.

"To account for possible drift of laser wavelength (e.g., the position of the absorption feature in a scan), a relationship between scan position and laser wavelength was estimated using a closely spaced pair of weak water absorption lines at 7281.72 and 7281.80 cm$^{-1}$ produced by a DFB laser-centered on a different wavelength than the one used for the measurements in this paper. The position of this pair was systematically scanned across the full temperature range of a single current ramp by slowly varying the setpoint of the WTC and the spacing between the two lines (0.08 cm$^{-1}$, or 0.015 nm) was measured in scan index (e.g., see Fig. 4). A linear fit to the ratio of this spacing to the difference in scan index as a function of scan position was determined as:

$$s(x)\,(nm/step) = 0.00052 + x*5.00*10^{-7}$$

where s is the change in wavelength per scan index (of the 445 points) and x is the scan index. The use of this function results in a near-constant line width as a function of wavelength if the position of the absorption feature shifts slightly due to variations in laser baseplate temperature. Although such a shift was never observed in these experiments, it is a consideration for use in an environment where the ambient temperature may vary significantly – e.g., by many tens of degrees. This method also allowed for the determination of the full width of the scan to be 0.279 nm for the specific scan start and end points and scan rate used in these experiments." (New L174-189)

L182: Accuracy metrics of the BMP280?

Old L182: We have added "with an accuracy of ± 1 % when compared to laboratory standards." (New L201)

L187: Does "These calculations" refer to the real time processing that is planned for future implementation and not the present version that is the focus of the manuscript?

Old L187: "These calculations take" has been changed to "Processing of spectra in real time takes". (New L207)

L190: The units of the x axis in Figure 4(b) are wavenumber, not wavelength as stated. Since wavelength is otherwise used consistently in the manuscript; I would suggest using wavelength here as well.

Old L190: Units in Fig. 4b have been changed to $cm^{-1}$ (i.e., wavelength). (New L210)

L190: In Figure 4(A), it might be helpful to use color on the trace to highlight the region of the scan used for the baseline fit. As noted in L132 comment, it would be clearer to plot a full scan including the 30 and 20 detector zero (laser off) points at the beginning and end of the scan.

Old L190: Points have been colored red in Figure 4a to show which points were used in the fit. (New L210)

L198: no hyphen needed between number and unit "25 L" even when used as an adjective.

Old L198: "25-L" changed to "25 L" (New L218)

L200: "saturated to a mixing ratio of ~27,000 ppm"—was the air in the chamber saturated (potential condensation)? Or was the saturation temperature of the generator lower than the ambient temperature? What is/are the values (uncertainty) of the mixing ratio reported by the reference CRDS measurement rather than "~"?

Old L200: The following description "The chamber was first saturated to a mixing ratio 200 of ~27,000 ppm with the dew point generator, after which lab air with ~13,000 ppm of H2O was

admitted to the chamber stepwise approximately every five minutes over the course of several hours" changed to:

> "A beaker of warm water was placed inside the chamber to humidify the air to a value just below the saturation point at lab temperature. Over the course of two hours, water vapor mixing ratios were reduced to ~13,000 ppm by stepwise-addition of relative dry ambient air from the laboratory into the chamber." (New L220)

L201: "admitted to the chamber"—also flow out of the chamber as well to maintain P?

Old L198: "a" has been changed to "an unsealed" (new L218)

L207: It would be good to include information about the linear regression as text in Figure 5

Old L211: the following has been added to the caption of Figure 5 caption (New L233):

> "Fit Parameters: slope = 0.0006, intercept = 0.0039, $R^2$ = 0.9999."

Old L209: remove period following "points)"; omit reference

Old L209: Period and reference have been deleted. (New L232)

L213: "Allan variance"

Old L213: The hyphen has been removed. (new 235)

L222: sensitivity is not affected by averaging— "detection limit"?

Old L222: "sensitivity" has been changed to "precision" (New L243)

L256: Omit "A long electrical line"? This was replaced with the "10 m twisted pair cable"? No comma needed after "cell" or hyphens between numbers and units; the word "long" could be omitted.

Old L256: This has been changed to "A 25 m fiber optic patch cable connected the output of the laser to the collimating lens on the input of the optical cell and a 10 m twisted pair of wires brought the detector signal back to the TDLS electronics box which was housed in the shipping container" (New L277)

L307: "Teensys"

Old L307: "Teensy's" has been changed to "Teensys" (New L328)

L320: "tested powered"

Old L325: "successfully tested powered" has been changed to "successfully powered" (New L346)

L336: "include"

Old L336: "including" has been changed to "include" (New L356)

L342: ABL already defined in introduction

Old L342: "atmospheric boundary layer (ABL)" changed to "ABL" (New L360)

In addition to the changes to address the referees' comments, we have made the following revisions:

A number of minor changes were made throughout text to correct typographical errors discovered in final proofreading.

Figure 2: The contrast was increased, and the acronym "GRIN" was deleted in the figure as it was incorrect.

Figure 4: The X-axis label has been changed to "Scan Index"

Figure 5: The units of the X axis have been changed to "$10^3$ ppm". The units of the Y-axis (top) have been changed to "$10^{-3}$ nm".

Figure 6: The Y-axis (top) changed to $10^3$ ppm

Figure 8: The units of Figure 8a have been simplified from "date/time" to "time" and the caption has been edited to clarify the starting date. The units have been changed to "$10^3$ ppm" on 8a and 8b.

Old L33-36: We have improved clarity by careful editing. New text reads:

"Observations of this variability are essential for elucidating the underlying micrometeorological processes and quantifying local-scale (100 m) radiation budgets important to the prediction of turbulent and convective processes and their impacts (Couvreux et al., 2009; Fabry, 2006; Ogunjemiyo et al., 2002). However, observations have been limited by the relatively high cost of existing instruments and the lack of high-quality data from more economical ones (Geerts et al., 2018)." (New L32-35)

Old L197: We have corrected the range over which calibration took place to 5000 to 25000 ppm to be consistent with Figure 5. (New L217-228)

Old L206: We added "This is larger than the precision of the Picarro, which is ~10 ppm, and so the deviation is mostly due to small differences in water vapor in the paths sampled by the two instruments." (New L228)

Old L311: We added part numbers for the optical cell and the manufacturer and part number for electronics box in the Table 1. (new L330)

---

## Author Response (AR3)

**Justification of Changes**

Note: Unless noted, line numbering refers to original document in AMT Discussions. "New" line numbering refers to the track-changes pdf version of the document.

**Per: Referee #1 (response in red)**

**General:**

The consistency of naming should be improved throughout the manuscript. Some names are changed multiple times as an example: Teensy = Teensy 4.1 = data collection microcontroller = receiver. Also introduced acronyms should be used or left out. More consistency here would be very helpful.

We have corrected the use of acronyms throughout. "Atmospheric Boundary Layer" has been defined once and used throughout the manuscript. "Tunable diode laser spectrometer" has been defined once and used throughout as the name of the developed instrument. Labeling of Teensy microcontrollers have been clarified.

The quality of language/sentence syntax should be improved. Many typos, slips of the pen, and complicated/convoluted sentences.

We have corrected a number of typos and some sentences to improve the quality of the paper. They appear in the track-changes version of the revised manuscript and will not be listed in detail here unless the changes were significant. We will summarize any additional significant changes at the end of this document.

The plotting ticks should be changed using to a more standard style as they are confusing that way. E.g.: Fig 5, yaxis highest tick should be 30, above the axis a common coefficient with  $1 \times 10^{-3}$ .

The format of the axes for Figure 5 (as well as Figures 6 and 8) has been changed, as recommended.

Also cross check references if missing and use correct citation style: Textual citation XXX et al. (YYYY) vs. parenthetical citations (XXX etl al. YYYY)

We have corrected citation style for the following in the text: L27: Santanello et al., 2018 moved to end of sentence. (New L29) L28: Larsen et al. 2002 has been removed. (New L29) L31-32: Fabry, 2006; Ogunjemiyo et al., 2002 moved to end of sentence. (New L37) Couvreux et al., 2009 has been added. (New L36-37) L41: Wulfmyer et. al 2015 moved to the end of sentence. (New L44-45) L55: Miloshevich et al., 2009 and 2004 reversed. (New L61-62) L58: "(Helbig et al., 2021; Petersen, 2016)" moved to end of sentence. (New L65) L65-66: Multiple references combined and placed at end of sentence. (New L74-75) L75: Multiple references changed to in-text citations. (New L84-85) L114: Rainwater 2022 moved to end of sentence. (New L135). L241: "(e.g. see Aslan et. al)" changed to "(Aslan et. al)". (New L294)

**2.1 Hardware Description**

The captions of Fig.1 to Fig. 3 should be extended to be self-explanatory and not reference to the text.

For example: Fig 1, please explain the different components shown: are drive input and TEC input circuits by themselves, how does the TIA look, are the laser driver and data collection unit the two Teensy microcontrollers...

Figure 1 caption (New L112-117) has been expanded to read the following:

"Schematic diagram of the new TDLS. Arrows represent the direction of information flow between individual components, including microcontrollers, laser, and temperature controller, or individual circuits, such as the transimpedance amplifier (TIA) and laser driver circuit. The components surrounded by the bold dashed line are contained on a single printed circuit board (schematic shown in Fig. 3). The output fiber from the laser is passed to the external optics through a FC/APC style fiber optic bulkhead coupler, and a twisted wire pair brings the detector signal back into the electronics box through a hermetic seal."

Figure 2 caption (New L155-159) has been expanded to include the following:

"Important elements of the TDLS laser scans as a function of time. The detector output (top panel) is the continuous voltage from the TIA. About one-third of the time the laser is powered off, and the signal is the background for the detector and TIA circuit. The laser drive (middle panel) represents the voltage output by the Teensy 3.6 used to set the current of the laser. A trigger pulse signal (bottom panel) sent by the Teensy 3.6 is read by the Teensy 4.1 to initiate sampling and recording of the scan."

L 103: The laser *wavelength* is tuned via temperature?

This has been changed to "The laser is tuned to the wavelength of a strong water absorption feature at 1368.59 nm by changing the temperature of a TEC in the laser butterfly package..." (New L119-120)

- L 105: add ± before 0.002 K "±" has been added. (new L122)
- L 127: adapt trigger pulse direction in Fig 1. From driver to data collection if the text is correct Figure 1 has been corrected to be consistent with the text. Text at new L145.

L129: After 5V reference to figure

The old Figure 2 has been replaced with a new figure, now numbered Figure 3, which is a circuit diagram of the complete electronics used in the instrument. The following sentence has been added (New L143):

"A complete electronic circuit diagram of the instrument is shown in Fig. 3."

L155 has been changed to

"Alternatively, it can be run indefinitely from a 7.5 V (or greater) DC power supply, as well as either of the Teensy microUSB 5V inputs." (New L180-181)

L 141 – 146: Please clarify: Was the laser collimated or divergent? How much bigger is the laser beamwidth compared to the InGaAs sensor active area? Have there been any tests regarding vibrations? (Even collimated laser beams show a distinct Gaussian profile which could induce signal variations upon vibration)

This has been changed to:

"The lens was configured so that the laser beam was divergent to fully illuminate the active area of a low-noise broadband indium gallium arsenide (InGaAs) semiconductor photodiode and reduce variations in intensity due to vibration and turbulent fluctuations of air density in the optical path. Several photodiodes from different manufacturers (FDGA05, Thorlabs; and FC1500, Fermionics, Simi Valley, CA) were used in this work at various times with no significant difference in results or performance." (New L164-170)

L 149: Syntax wrong or double sentence Duplicate wording in L149 has been removed. (New L173)

**2.2 Spectral Processing**

L 163: please cross check values on where the offset/drift is determined 30 & 20 points vs. 10 point in fig 4 caption.

This has been changed to:

"Briefly, a small detector/amplifier offset is determined from 10 points at the start and 10 points at the end of each scan while the laser is powered off." (New L192-193)

The following sentence has been added to Figure 4 caption:

"The fit is made between the points highlighted in red (30 points at the start of the scan and 20 points at the end)." (New L239-240)

L 165 – 169: This part requires more clarification on how to convert scan steps into wavelength. The following paragraph has been added:

"To account for possible drift of laser wavelength (e.g., the position of the absorption feature in a scan), the relationship between scan position and laser wavelength was estimated using a pair of closely spaced water absorption lines at 1373.3002 and 1373.2878 nm emitted by a similar model DFB laser centered on a different wavelength than the one used for the measurements in this paper. The position of this pair was systematically scanned across the full temperature range of a single current ramp by slowly varying the setpoint of the laser TEC temperature controller, and the spacing between the two lines (i.e.,

 $\Delta\lambda$ =0.0124 nm) was determined in units of scan index (e.g., see Fig. 4). A linear fit to the ratio of this spacing to the difference in scan index was determined as a function of scan position:

 $s(x) (\Delta nm / \Delta step) = 0.00052 + x * 5.00 * 10^{-7}$

where s(x) is the change in wavelength per scan index (of the 445 points) and x is the scan index value. Using this function results in a near-constant line width as a function of wavelength if the position of the absorption feature shifts due to variations in laser baseplate temperature. Although such a shift was never observed in these experiments, it is a consideration for measurements in an environment where ambient temperature may vary significantly (e.g., by many tens of degrees). This method also allowed for the determination of the full width of the scan to be 0.279 nm for the specific scan start and end points and scan rate used in these experiments. (New L196-214)

L 170: What do you want to say with the "[..] are then placed in an array? We decided that this level of detail is unnecessary and so the following was deleted:

"The observed signal (i.e., Iobs(t)) and calculated background I0(t) are then placed in an array [i, Io(i), I(i)]." (New L214).

Fig4 (b) Unit should be wavenumber

The x-axis units in Figure 4b are now wavelength in nm, and we do not use wavenumbers in any part of the manuscript.

**3. Results:**

Fig 5: Does the x-axis represent values measured by the Picarro or converted values from the TDLS? Please also state the lin-regress function parameters (slope, intercept,  $r^2$ ) or a plot of the converted H2O-ppm from TDLS over H2O-ppm measured by the Picarro together with a 1:1 line and respective regression slope. Similar as for Figure 8 (b)

Fig. 5 x-axis label changed from "H2O mixing ratio (ppm)" to "Picarro H2O mixing ratio (ppmv)" and the following has been added to the caption of Figure 5 caption:

"Fit Parameters: slope = 0.0006, intercept = 0.0039, R2 = 0.9999." (New L263-L264)

L 216 - 220: This part needs more clarification. Was a different InGaAs sensor used for the calibration of the instrument than the actual measurement? That could yield a different conversion coefficient.

The following has been added earlier in the manuscript:

"Several photodiodes from different manufacturers (FDGA05, Thorlabs; and FC1500, Fermionics, Simi Valley, CA) were used in this work at various times with no significant difference in results or performance." (New L168-170)

- L 239: Wu et al 2015 citation does not present in references Wu et al. 2015 has been added to the references. (New L292)
- Fig8 (a). Please convert the x-axis to actual time in UTC Figure 8 x-axis has been converted to UTC.

L 278: Doesn't the averaging over 30s smooth all variation on a spatial scale of 1.5m? Hence I am not surprised that that values align even the locations had a 1.5m separation.

Old L281: The following has been added.

"...including variability due to the CRDS inlet and optical cell being separated by 1.5 m." (New L335-336)

**Some Typos**

- L 104: remove dot after "([...], MT)"
- L 111: add quotes " after receiver
- L 122: resister -> resistor
- L209: remove dot after "(black points)"
- L 256: lowercase a before 25-m long
- L 298: Remove dot after power
- L 320: mission word(s) after water
- L 325: Either tested or powered
- L 333: remove with before TEC
- L 337 delete "configuring for use"
- L 342: ABL was already introduced in the introduction These have all been corrected in the revised manuscript, and are not listed individually here.

**Per Referee #2:**

**Specific comments:**

L8: First sentence could be restructured to remove redundancy of "high spatiotemporal variability" and "abundances varying..."

"The high spatial temporal variability of" has been deleted, and the sentence now reads:

"Water vapor in the atmospheric boundary layer poses a significant measurement challenge with abundances varying by an order of magnitude over short spatial and temporal scales." (New L8)

ABL should be defined here rather than L11, and then L11 could just use "ABL"

"Atmospheric boundary layer" appears first on New L8 in the abstract and again on New L26 in the Introduction, and the abbreviation "ABL" is used for the remainder of the text.

L8: "possesses" should be "poses" (or "presents") "possesses" changed to "poses". (New L8)

L11: "in situ" is a Latin term and not hyphenated. I would think that you would want to include "open-path" in the description here since that is a critical aspect enabling the fast response time.

"Open-path added", and "in situ" corrected. (New L10 and L11)

L11: "tunable diode" is generally not hyphenated (although I see it also was in Dorsi et al 2014).

**"Hyphen deleted"**

L11: You define 'TDLS' first in the abstract as "tunable diode laser spectroscopy", but then use it and subsequently define it as "tunable diode laser spectrometer". Perhaps use "tunable diode laser

absorption spectroscopy" in the abstract since that is the technique utilized and then TDLS as the spectrometer.

The instance where "TDLS" was used to refer to the method has been removed (New L12), and we retain the abbreviation of the instrument as "TDLS" throughout.

L12: you only need to include the acronym definition here if you will use the acronym alone later in the abstract. Comment also applies to L16 and L21.

"(SWIR)" is deleted as it is not used in other locations in text. (New L12)

- L16: "proportional integral", as appears in L104 of the text "proportional-integrating" is changed to "proportional-integral" (New L16-17)
- L17: "comprised of" should technically be "composed of" or "comprises" "comprised of" is changed to "constructed from" (New L18)
- L18: perhaps "agreed" "agrees" is changed to "agreed" (New L19)

L19: perhaps "will allow" and preface with something like "The instrument is robust and simple to operate"

"allows" is changed to "is robust and simple to use and will" (New L20)

L29: "tropics"

"Tropics" is changed to "tropics" (New L30)

L32: "underlying mesoscale processes"—meteorologically, mesoscale is typically 10 to 100s of km, which doesn't seem appropriate here

"mesoscale meteorological" changed to "micrometeorological" (New L34)

- L39: "DIALs and Raman lidars" or "differential absorption and Raman lidars" "DIAL and Raman lidars" changed to "differential absorption LIDARs and Raman LIDARs." (New L43)
- L45: "such as infrared gas analyzers (IRGAs)" "the" removed (New L49)
- L46: "have come to" -> "are typically used to" Change is made (New L49)

L47: it is really the cost (~x10, and for some applications the size/weight), not the limited number of vendors or some "highly specialized" nature that is the limitation, right? And potentially differences in required maintenance/recalibration? You expect the new TDLS to not require recalibration (does require initial calibration per L177) or regular maintenance, correct?

Text has been added or edited to address this point:

"These research-grade instruments, which are used predominantly at multi-instrumented flux towers and weather stations and tend to be expensive, often costing \$20,000 or more. In addition, they can incur additional costs for factory service to maintain high accuracy. Consequently, their use in remote locations has been relatively limited." (New L50- 53)

"High-resolution in situ observations of H2O are essential for numerical weather prediction and for investigations of the evolution of the ABL and its turbulence characteristics (e.g. large eddy simulations), and there is a need for more frequent measurements from remote locations (Helbig et al., 2021; Petersen, 2016). We report here on the development of an economical new fast-response laser spectrometer. The instrument is capable of highresolution measurements of water vapor in the ABL while demonstrating high accuracy and precision comparable to that of commercially available research-grade instruments. Built from low-cost components that are readily available commercially, the instrument exhibits relatively low up-front costs with the ability to replace critical components, thus bridging the gap between the more expensive and highly accurate fast-response instruments and the relatively inexpensive, but slower response capacitive instruments." (New L63-72)

L57: "prediction"

L57: "capable of"

L60: "development and performance"?

L61: "high accuracy and precision matching that of" and "lower cost and greater flexibility that would allow widespread deployment for routine observations"

These changes are included in the new revised wording noted above (New L63-72)

L67: "laser diode"? and what is meant by a "generic" package since it does require built-in TEC and tight coupling of the fiber?

"generic" replaced with "common butterfly" (New L76)

L69: "components"? and "components and exhibits" "technology" is changed to "components" (New L78)

L71: I'm a little skeptical of the emphasis on the use of the instrument by fully inexperienced operators.

"with research grade instruments" is changed to "in laser spectroscopy" (New L81)

L76: "2023), the reported instruments have had a slow response, resulting in limited vertical resolution"

This is changed to "the instruments used have slow response times, resulting in limited vertical resolution" (New L85)

**L78: an example of a location?**

We have added "remote land and ocean regions" and include a new citation to J. Brotzge, et al., 2023, a reference we have added to our list (New L88)

**L81: what is meant by "terrain and variable inhomogeneity"?**

At the recommendation of the Editor, this has been clarified to "large spatial and temporal gradients in humidity due to adjacent complex terrain that contributes to significant errors in latent heat fluxes derived from those measurements" (New L90-92)

L89: "based on" would be more appropriate This is changed to "based on" (New L100)

L90: the clause "a schematic of which is shown in Fig. 1." currently references the previously reported (Dorsi et al 2014) instrument. The clause could be inserted immediately after "described here" in L89 to be clear.

We have changed to "An overview of the instrument is depicted in Fig. 1." (New L108)

L92: "is rapidly scanned"; "variations, a short" Changes are made. (New L103)

"(NLK1E56AA, NTT Innovative Devices, Yokohama, Japan)" has been moved to New

"(NLK1E56AA, NTT Innovative Devices, Yokohama, Japan)" has been moved to New L101-2 to improve clarity.

L100: Figure 1 shows the trigger pulse passing from the receiver microcontroller to the laser drive, but the text states that the trigger pulse for data collection originates from the laser driver board.

Figure 1 has been corrected to be consistent with the trigger signal described in the text.

L104: "TEC controller" This change is made. (New L120)

L105: "temperature of 0.002K" should be "temperature of XX.XXX ± 0.002 K" or say "A temperature stability of ±0.002 K, consistent..." "±" has been added (New L122)

L107: "DFB" should be "laser" (or "DFB laser diode") This is changed to "laser" (New L125 and L134)

L108: "a digital-to-analog (DAC) output" since the 3.6 has two, although the 4.1 does not have a DAC, so only from the 3.6 (although, as noted, now discontinued).

- This sentence is changed to "If desired, a voltage from a digital-to-analog (DAC) output can be used for dynamic temperature control." (New L125)
- L110: "Arduino-compatible" hyphenated? But not "laser driving" or "data acquisition" As recommended by the Editor, this sentence has been clarified. It now reads: "Two independent Arduino-compatible microcontrollers were chosen for separately driving the laser (a Teensy 3.6) and for data acquisition (a Teensy 4.1)."(New L127)

L111: "based on"

This is changed to "employ" (New L129)

L112: "previous instruments" developed in your lab? Or universally? We have changed to "developed in our lab that employ the same measurement technique as reported here" (New L130-L131)

L117: "scans to ~10 kHz and faster, resulting in high precision of the measurements" precision from averaging over multiple scans? Current operation is only 10 Hz (100 msec) scans? L320 says "tests showing that full scans over the water [line] at ~1000 Hz are possible" and that higher scan (measurement) rates result in reduced precision (for individual scans)

This has been changed to "up to 1 kHz" and "resulting in high-precision of the measurements" has been deleted. (New L136)

L120: Reference to Figure 2 is missing from the text (~L129?). Fig 3 is already mentioned on L126. Reorder sentences to put "Prior to..." after the circuit discussion? Would it make sense to

include Figure 2 in supplemental material? That would allow additional inclusion of the custom TIA circuit and supporting circuit board.

Based on this and the Editor's comments, we now include a circuit diagram of the entire instrument, now numbered Figure 3. The following has been added:

"The middle panel in Fig. 2 shows an example of a series of linear ramps used as the drive function, each consisting of 1366 discreet one-bit steps from 0.80 V to 1.9 V. This voltage is conditioned with an operational amplifier (LT1101, Analog Devices, Wilmington, MA) that controls the current required to scan the laser from a transistor (TIP 32AG n-channel transistor) in a textbook voltage-to-current converter circuit (Figure 6.31 of Horowitz and Hill, 1983). A complete electronics circuit diagram is shown in Fig. 3. The scan rate, current range, and a pause for background time are configured in software." (New L139-144)

We added Horowitz and Hill, 1983 to the reference list.

We have also added "The top panel in Fig. 2 shows the continuous output of this circuit." (New L173-174).

L129: "A Teensy model 4.1 with a built-in Micro-SD card feature was used..."; "a trigger pulse"

L131: ADC not defined at first use; "data acquisition analog-to-digital conversion (ADC) is started."?

L132: There is some discrepancy regarding the discussion of Fig 3. It says here that the plot contains 445 points, but the figure shows 4 complete scans. Figure 4 shows 445 points without showing a complete scan ( $\sim 10+425+\sim 10$ ?). It would be best to clearly describe the sequence of one scan (475 points? 30 + 425 + 20?) and show the complete scan in Figure 4.

This paragraph has been edited for clarity. It now reads:

"Before the start of each scan, the Teensy 3.6 produces a voltage pulse ("trigger"), shown on the bottom panel of Fig. 2, that initiates the data acquisition and storage process on the Teensy 4.1. At this time, the internal clock is recorded into a buffer, and the output from the detector TIA is recorded onto a MicroSD card as a single scan consisting of 445 discreet samples at 12-bit resolution. Although the Teensy 4.1 samples at 300 ksps, we oversampled 32 times using a software function that reduces noise inherent in the analog-to-digital converter (ADC)." (New L145-151)

L132: How does the math for 7.2 kHz "raw" ADC work with 475 pts / 100 msec at 32x oversampling? Does 7.2 kHz already include the 32x and so is faster than the 4750 samples / sec?

The change described above should address this comment.

L144: omit "on the opposite side of optical path both operated in photovoltaic mode"? A following sentence begins "The photodiode is operated in photovoltaic mode"

The phrase "(either Thorlabs FDGA05 or Fermionics FD1500) on the opposite side of optical path both operated in photovoltaic mode" has been deleted. (New L165-L166)

L148: It seems like the "AD1101, Analog Devices" is actually "HMCAD1101"? I could not find a part at Analog Devices that was just "AD1101".

This is changed to "LT1013" (New L172)

L165: description here is "1st-order polynomial" while the caption in Fig 4 uses "linear fit" these are indeed the same thing, but it might be clearer to be consistent. This is changed to "linear" (New L194)

L166: It would be useful to have a little more clarity on the process of converting the temperature – wavelength determination to the current ramp scan to account "for the possible drift of the tune temperature by removing the nonlinear output laser wavelength response to a linear current ramp" and determination of the scan wavelength range.

New L196-214 was added to address this point (see response to Reviewer 1).

L182: Accuracy metrics of the BMP280?

We have added "with an accuracy of  $\pm 1$  % when compared to laboratory standards." (New L226-7)

L187: Does "These calculations" refer to the real time processing that is planned for future implementation and not the present version that is the focus of the manuscript?

"These calculations take" changed to "Processing of spectra in real time takes" (New L232)

L190: The units of the x axis in Figure 4(b) are wavenumber, not wavelength as stated. Since wavelength is otherwise used consistently in the manuscript; I would suggest using wavelength here as well.

Units in Fig. 4b have been changed to nm (i.e., wavelength)

L190: In Figure 4(A), it might be helpful to use color on the trace to highlight the region of the scan used for the baseline fit. As noted in L132 comment, it would be clearer to plot a full scan including the 30 and 20 detector zero (laser off) points at the beginning and end of the scan.

"Points are colored red in Figure 4a to show which were used in the fit." added (New L239-40)

L198: no hyphen needed between number and unit "25 L" even when used as an adjective. Hyphen removed (New L244)

L200: "saturated to a mixing ratio of ~27,000 ppm"—was the air in the chamber saturated (potential condensation)? Or was the saturation temperature of the generator lower than the ambient temperature? What is/are the values (uncertainty) of the mixing ratio reported by the reference CRDS measurement rather than "~"?

This description now reads:

"The TDLS integrals were calibrated by sampling a range of mixing ratios in an unsealed 250 L Polycarbonate chamber from 6,970 ppmv to 25,700 ppmv as reported by a Picarro CRDS. The TDLS optical cell was placed in the center of the chamber, and a fan was used to ensure the chamber was well-mixed. The sampling line of the CRDS was aligned with the mid-point of the TDLS open-path cell and positioned just outside the path of the laser beam. A beaker containing warm water was placed inside the chamber to humidify the air to a value just below the saturation point at lab temperature. Over the next two hours, mixing ratios were reduced to 13,520 pmv by stepwise addition of relatively dry ambient air from the laboratory into the chamber. Values below 13,000 ppmv were produced by further dilutions using a flow of dry air

from a cylinder of Ultra Zero Air (H2O < 2 ppm, total hydrocarbons < 0.1 ppm, Airgas, Dacono, CO)." (New L244-54)

- L201: "admitted to the chamber"—also flow out of the chamber as well to maintain P? We now specifically state the chamber is unsealed (New L244)
- L207: It would be good to include information about the linear regression as text in Figure 5 The fit parameters (slope = 0.0006, intercept = 0.0039,  $R^2$  = 0.9999) are now listed in Figure 5 caption. (New L263-4)

L209: remove period following "points)"; omit reference

- L213: "Allan variance"
- L222: sensitivity is not affected by averaging— "detection limit"? Period and reference have been deleted. (New L261-2) The hyphen has been removed. (New 265-6) "sensitivity" has been changed to "precision" (New L275)

L256: Omit "A long electrical line"? This was replaced with the "10 m twisted pair cable"? No comma needed after "cell" or hyphens between numbers and units; the word "long" could be omitted.

This sentence now reads:

"A 25 m fiber optic patch cable connected the output of the laser to the collimating lens on the input of the optical cell and a 10 m twisted pair of wires brought the detector signal back to the TDLS electronics box which was housed in the shipping container." (New L310-312)

L307: "Teensys"

L320: "tested powered"

L336: "include"

L342: ABL already defined in introduction Now "Teensys" (New L362) Changed to "successfully powered" (New L381) Changed to "include" (New L393) Changed to "ABL" (New L398)

In addition to the changes to address the referees' comments, we have made the following revisions:

Minor changes were made throughout text to correct typographical errors discovered in final proofreading.

Figure 1 caption modified to: "The components surrounded by the dashed line are contained on a single printed circuit board (schematic shown in Fig. 3)."

Figure 2: The contrast was increased, and the acronym "GRIN" was deleted in the figure as it was incorrect.

Figure 4: The X-axis label has been changed to "Scan Index"

Figure 5: The units of the X axis have been changed to " $10^3$  ppm". The units of the Y-axis (top) have been changed to " $10^{-3}$  nm". The Y axis on the top panel was rescaled to reveal

one data point that was missing in the original figure.

Figure 6: The Y-axis (top) changed to  $10^3$  ppm

Figure 8: The units of Figure 8a have been simplified from "date/time" to "time" and the caption has been edited to clarify the starting date. The units have been changed to " $10^3$  ppm" on 8a and 8b.

L33-36, L50-53, and L57-60: We have improved clarity by careful editing. New text reads:

"Observations of this variability are essential for elucidating the underlying micrometeorological processes and quantifying local-scale (100 m) radiation budgets important to the prediction of turbulent and convective processes and their impacts (Couvreux et al., 2009; Fabry, 2006; Ogunjemiyo et al., 2002). However, observations have been limited by the relatively high cost of existing instruments and the lack of high-quality data from more economical ones (Geerts et al., 2018)." (New L33-38)

"At the other end of the cost spectrum are various versions of capacitive humidity sensors that employ thin-film water-sensitive polymers sandwiched between two electrodes." (New L56-7)

"High-resolution in situ observations of  $H_2O$  are essential for numerical weather prediction and for investigations of the evolution of the ABL and its turbulence characteristics (e.g. large eddy simulations), and there is a need for more frequent measurements from remote locations (Helbig et al., 2021; Petersen, 2016)." (New L63-65)

"There is a need to increase the density of measurements on specific reservoirs to map out the large spatial and temporal gradients in humidity due to adjacent complex terrain that contributes to significant errors in latent heat fluxes derived from those measurements (Friedrich et al., 2018)." (New L89-93)

**L206: We added:**

"This is larger than the precision of the CRDS, which is  $\sim 10$  ppmv, and so the deviation is mostly due to small differences in water vapor in the paths sampled by the two instruments. (New L257-9)

L311: In Table 1 we added part numbers for the optical cell and the manufacturer and part number for electronics box. (New L365)